# Single-nuclei and bulk-tissue gene-expression analysis of pheochromocytoma and paraganglioma links disease subtypes with tumor microenvironment

Pheochromocytomas (PC) and paragangliomas (PG) are rare neuroendocrine tumors associated with autonomic nerves. Here we use single-nuclei RNA-seq and bulk-tissue gene-expression data to characterize the cellular composition of PCPG and normal adrenal tissues, refine tumor gene-expression subtypes and make clinical and genotypic associations. We confirm seven PCPG gene-expression subtypes with significant genotype and clinical associations. Tumors with mutations in *VHL*, SDH-encoding genes (*SDHx*) or *MAML3*-fusions are characterized by hypoxia-inducible factor signaling and neoangiogenesis. PCPG have few infiltrating lymphocytes but abundant macrophages. While neoplastic cells transcriptionally resemble mature chromaffin cells, early chromaffin and neuroblast markers are also features of some PCPG subtypes. The gene-expression profile of metastatic *SDHx*-related PCPG indicates these tumors have elevated cellular proliferation and a lower number of non-neoplastic Schwann-cell-like cells, while *GPR139* is a potential theranostic target. Our findings therefore clarify the diverse transcriptional programs and cellular composition of PCPG and identify biomarkers of potential clinical significance.

Pheochromocytomas (PC) and paragangliomas (PG) are rare neuroendocrine tumors of the adrenal medulla or extra-adrenal paraganglia, respectively. Most PG arise in the distribution of paraxial sympathetic nerves. A subset, referred to as "head and neck PG" (HN-PG) or "parasympathetic PG" arise in paraganglia associated with the vagus or glossopharyngeal nerves, e.g., from the oxygen sensing chief cells (glomus cells) of the carotid body[1]. Sympathetic PCPG arising from chromaffin tissues release "fight or flight" hormones (e.g., epinephrine or norepinephrine) causing symptoms including hypertension, sweating, palpitations, headache as well as severe cardiovascular conditions if left untreated[2,3]. Although all PCPG have metastatic potential[4], metastases only arise in 10–20% of patients[5]. There is a clinical need for biomarkers that can predict metastatic progression in patients. Treatments for metastatic PCPG include surgery, radiation,

chemotherapy, radiopharmaceuticals and tyrosine kinase inhibitors[6] but none are curative with a median survival of ~6 years[7]. Improved treatments will rely upon a better understanding of molecular and cellular disease characteristics.

PCPG are heterogeneous reflected by their genetic, transcriptional, DNA methylation, biochemical and metabolic profiles[8–13]. Up to 40% of PCPG are hereditary involving more than 20 PCPG susceptibility genes, whereas sporadic disease caused by somatic mutations or gene-fusions account for an additional 30–40% of cases[14,15]. Early microarray analysis clustered PCPG into two groups; so-called cluster 1 (C1) and cluster 2 (C2)[9] with further division later described[10]. C1 PCPG are noradrenergic and pseudohypoxic, involving stabilization of hypoxia-inducible factors (HIFs) under normoxic conditions[8]. Genes implicated in C1 PCPG include *EPAS1* (encoding HIF-2α subunit), HIF

e-mail: rtothill@unimelb.edu.au

regulators including *VHL* and *EGLN1* (encoding PHD2), and Krebs cycle genes such as succinate dehydrogenase subunits (*SDHA-D*) and *FH* among others[16,17]. These mutations in the Krebs cycle cause accumulation of metabolic intermediates (e.g., succinate, fumarate) and inhibition of the 2-oxoglutarate-dependent dioxygenases including HIF-regulating prolyl hydroxylases and DNA and histone demethylases[18,19]. C2 tumors are well-differentiated adrenergic tumors typified by germline or somatic mutations in kinase signaling genes (e.g., *RET*, *NF1*, *HRAS*), *TMEM127,* and the MYC-binding partner *MAX*[10,15,20]. An additional "WNT-signaling" cluster associated with somatic *MAML3*-fusions was more recently defined[15]. PCPG genotype is associated with disease penetrance, metastatic risk and theranostic targets[21].

In addition to neoplastic cells per se, stromal and immune cells in PCPG are likely to be important for understanding tumor biology and treatment[22]. Pseudohypoxia drives a proangiogenic program and C1 tumors are highly vascularized with irregular vascular patterns[23]. Outside of histomorphology, inference from bulk RNA gene-expression analysis and staining of select proteins in tissues, relatively little is known of normal cell types within these neoplasms. Furthermore, analysis of immune cells across a large series of PCPG has been largely restricted to in silico predictions from bulk RNA data in the context of pan-cancer analysis[24].

Herein, we apply single-nuclei RNA-seq (snRNA-seq) to a broad selection of PCPG genotypes to identify stromal and immune cell composition of tumors, compare the transcriptional profiles of neoplastic and normal cells within tumors and normal adrenal tissues, respectively, as well as infer cell–cell signaling. Furthermore, a large compendium of bulk-tissue gene-expression data is used to confirm tumor genotype-subtype associations, validate the relative expression of cell type markers across PCPG gene-expression subtypes and identify genes and gene-sets differentially expressed between metastatic and non-metastatic PCPG.

## Results

### Application of single-nuclei RNA-seq to frozen PCPG tumors

To explore PCPG and normal adrenal medulla (NAM) at single cell resolution we applied droplet-based snRNA-seq to 32 frozen tumors or healthy normal adrenal medulla (NAM) tissues (Fig. 1a) (see Methods). Samples included 18 PC and seven abdomino-thoracic PG (AT-PG), five HN-PG as well as two NAM tissues (Fig. 1b). PCPG represented 13 known somatic or germline PCPG driver genes and were all primary tumors (Supplementary Data 1). Three tumors were from patients who developed metastatic disease (E205, PC, *EPAS1*; E206, PC, *EPAS1*; E235, PC, *MAML3*-fusion), and two tumors had locally invasive features (E007, *TMEM127*; E025, AT-PG, *SDHA*). Two tumors were synchronous PG from the same patient (P018, PGL1 and PGL3, *SDHB*).

After quality filtering (see Methods and Supplementary Fig. 1), 109,238 nuclei from 32 samples were clustered by UMAP (Fig. 1c). Stromal and immune cell nuclei clustered by cell lineage, indicating minimal technical variability, whereas neoplastic (NEO) nuclei clustered by patient sample (Fig. 1c, d). Annotation of samples by processing batch did not show obvious batch effects that would explain the UMAP clustering pattern (Supplementary Fig. 2). Applying batch correction by sample (see Methods) resulted in co-clustering of NEO cells (Supplementary Fig. 3); however, it also caused removal of the expected transcriptional differences between PCPG genotypes. Therefore, uncorrected data were used for downstream analysis.

Stromal and immune cells were identified by similarity to the FANTOM5 reference (see Methods) and cell type marker genes (Fig. 1e). Broad cell lineages included fibroblasts, endothelial cells (ECs), Schwann cell-like cells (SCLCs), myeloid cells, mast cells, T/NK cells, B cells, and adrenocortical cells. The contribution of stromal and immune cells within tumors was variable, from 0.5 to 76.7% of total nuclei (Fig. 1f, Supplementary Data 2). A more detailed analysis of normal cells is described below. NEO and normal chromaffin cells were

identified by *TH* and *CGHA* expression (Fig. 1e). NEO cells from parasympathetic HN-PG expressed *NRG3* and *LEF1*, the latter identified as a cancer checkpoint in HN-PG[25]. Low-level detection of chromaffin-related mRNA transcripts in unrelated cell types may be explained by ambient RNA; a known technical artefact with droplet-based protocols. Attempts to remove this artefact bioinformatically were unsuccessful therefore careful inspection of genes of interest was done to confirm cell type specificity (see Methods).

NEO cells were confirmed by inference of aneuploidy (InferCNV), showing chromosomal loss profiles concordant with SNP-array data (where available) and loss of heterozygosity for PCPG tumor-suppressor driver genes including *VHL* (chr3p), *NF1* (chr17q), *TMEM127* (chr2q) and *SDHB* (chr1p) (Supplementary Fig. 4). Sub-clonal NEO cell populations with additional chromosomal changes were found in some PCPG tumors (e.g., P018-PGL1, P018-PGL3, E210, E209, E196, E208) (Supplementary Fig. 4). For instance, P018-PGL1 involved sub-clonal ch2q loss previously detected by SNP-array[26] and ~25% of cells had this alteration by snRNA-seq (Fig. 1g).

UMAP clustering of NEO cells was broadly concordant with genotype or PCPG subtypes (Fig. 1h). For instance, NEO cells from tumors with driver genes involving *MAX*, *MAML3*-fusion, *NF1*, and *SDHD* generally co-clustered by their respective genotype, whereas two *FH*-mutant PC had greater interpatient heterogeneity (Supplementary Fig. 5). Interestingly, for two synchronous primary PG (P018 - PGL1 and PGL3) we previously reported evolutionary convergence based on similar chromosomal copy-number alterations[26]. NEO cells from PGL1 and PGL3 co-clustered tightly by UMAP, indicating these synchronous tumors also had highly similar transcriptional profiles (Supplementary Fig. 5).

### Integration of snRNA-seq with bulk-tissue gene expression data validates PCPG subtypes

Despite the utility of snRNA-seq to identify genotype associations among PCPG, the limited number of biological replicates among genotypes limited the generalizability to the broader PCPG population. We therefore sought to integrate our snRNA-seq data with a large compendium of published microarray and RNA-seq data (*n* = 735 samples) (Supplementary Data 3). Following the removal of poor-quality samples, data were harmonized to remove platform and dataset biases (see Methods and Supplementary Fig. 6). UMAP was used to cluster combined bulk-tissue and "pseudo-bulk" snRNA-seq data (pooled total cells or NEO cells only) (Fig. 2a). No major biases were observed by study or data type (Supplementary Fig. 6). For 14 of 32 snRNA-seq profiled samples, matched bulk-tissue RNA data was available. Among paired data, pseudo-bulk snRNA-seq and bulk-tissue samples co-clustered, whereas discordant pairs were resolved by clustering pseudo-bulk NEO cells only (removing the contribution of normal cells), for example, PC tumors E018 (*RET*) and E209 (*MAX*), respectively (Fig. 2b).

UMAP clustering showed significant associations with PCPG driver genes (Fig. 2a, d Supplementary Data 4). Consensus clustering identified nine PCPG subtypes, but two consensus clusters were consolidated based on their UMAP proximity and their association with the kinase signaling pathway (Fig. 2c). Two bulk-tissue datasets were previously used for subtype discovery (COMETE and TCGA)[10,15], enabling a comparison of PCPG subtype annotations (Fig. 2e). Subtypes were in broad agreement, but increased resolution in our analysis was observed. For continuity we adopted the C1/C2 nomenclature and assigned an index PCPG driver gene or pathway to each subtype: $C1A_1$ (*SDHx*), $C1A_2$ (*SDHx*-HN), $C1B_1$ (*VHL*), $C1B_2$ (*EPAS1*), C2A (Kinase), $C2B_1$ (*MAX*), and $C2B_2$ (*MAML3*).

The C1 or "Pseudohypoxia" group was separated into four subtypes. Based on available clinical annotation, $C1A_1$ (*SDHx*) tumors were predominantly AT-PG (69%) and were enriched for *SDHB* and *SDHD* genotypes (Fisher's test Benjamini-Hochberg (BH) adj. *P*-value <

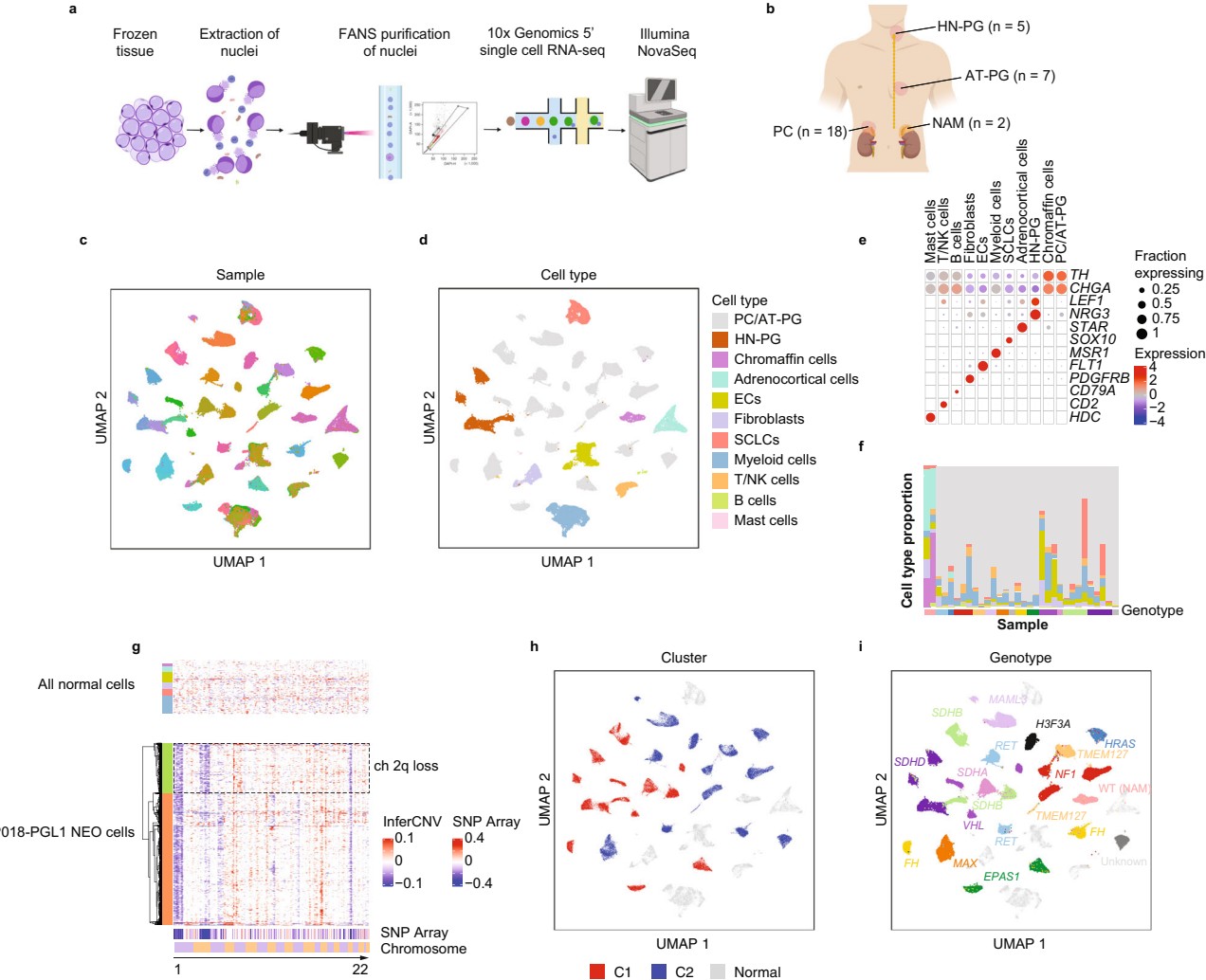

**Fig. 1 | snRNA-seq for cell type identification, inferred copy-number analysis and clustering of 30 PCPG and two NAM tissues. a** Schematic describing the workflow for single-nuclei isolation, Fluorescence-Activated Nuclei Sorting (FANS) and 5′ chemistry using the 10x platform starting from frozen tissues (created with BioRender.com). **b** Anatomical locations and sample size of tumors profiled with snRNA-seq (created with BioRender.com). **c** UMAP clustering of all nuclei from all samples colored by sample (individual color-sample key not shown, nSamples = 32). **d** UMAP clustering of all nuclei from all samples colored by cell type. ECs: Endothelial cells, SCLCs: Schwann cell-like cells. (nNuclei = 109,238). **e** Expression of major cell type markers in annotated UMAP clusters. Expression scale = Z-score standard deviations from mean. **f** Relative proportion of cell types detected in individual samples (cell type colored as per panel **d**). **g** Inferred copy-number in NEO and non-NEO cell types by gene-expression for an *SDHB*-associated AT-PG (P018-PGL3) (bottom panel; tumor cells with tumor subclones colored orange and green), top panel; non-neoplastic cell types from all tumors and NAM tissues with cell type colored as per panel **d**. InferCNV scale = modified expression, SNP array scale = log ratio. **h** UMAP clustering of all NEO nuclei from all samples colored and labeled by their known associated PCPG gene-expression subtype (C1/C2) (nNuclei = 109,238). **i** UMAP clustering with NEO nuclei labeled and colored by PCPG genotype (nNuclei = 109,238).

0.001). C1A₂ (*SDHx*-HN) included all HN-PG tumors in the bulk dataset although HN-PG were still only the minority (26%, 8/31) of tumors in this group. C1A₂ (*SDHx*-HN) were also enriched in *SDHD*-mutants (Fisher's test BH-adj. *P*-value < 0.001). Clinical annotation showed an enrichment of metastatic PCPG in both C1A₁ (*SDHx*) and C1A₂ (*SDHx*-HN) subtypes (Fisher's exact test BH-adj. *P*-value < 0.001, Supplementary Table 1).

The COMETE C1B cluster was split into C1B₁ (*VHL*) and C1B₂ (*EPAS1*), the latter enriched for *EPAS1*-mutants (Fisher's test BH adj. *p* < 0.001), consistent with *EPAS1*-mutant PCPG having a distinct gene-expression profile[27]. One *FH*-mutant PCPG (E211) clustered to C1B₂ (*EPAS1*) and the other (E210) to C2B₂ (*MAX*). In case E211, the *FH* germline variant (*FH* c.700 A > G) has previously been associated with FH-deficient PC[28] and loss of chr1q (including the *FH* locus) was inferred by snRNA-seq (Supplementary Fig. 7). The *FH* splice-site variant (*FH*, c.268-2A) in case E210 has also been reported as a pathogenic variant[17]. Evidence of altered *FH* RNA-splicing was found in E210

involving loss of exon 5 expression 3-prime of the *FH* variant, although arm level chr1q loss was not found by snRNA-seq (Supplementary Fig. 7). The observed subtype clustering of *FH*-mutants was unexpected given *FH*-deficient PCPG are thought to share molecular and metabolic features with *SDHx*-deficient PCPG[29]. Our data suggest that *FH*-deficient PCPG may be more heterogeneous than previously described, although the involvement of other driver genes cannot be excluded in these tumors.

Consistent with prior studies, the C2A (Kinase) subtype was associated with genes involving kinase signaling (e.g., *NF1*, *RET*, *HRAS*, *TMEM127*). C2B (COMETE) or WNT-altered (TCGA) subtypes were overlapping, but in our analysis these tumors partitioned into two groups; the heterogeneous C2B₁(*MAX*) subtype consisting of *MAX*, *CSDE1*, and *H3F3A* mutants and many with no reported driver; and C2B₂ (*MAML3*) consisting of 10 of 11 *MAML3* fusion-positive PC. The C2B₂ (*MAML3*) subtype was positively associated with metastatic disease, consistent with the WNT-altered (TCGA) subtype having

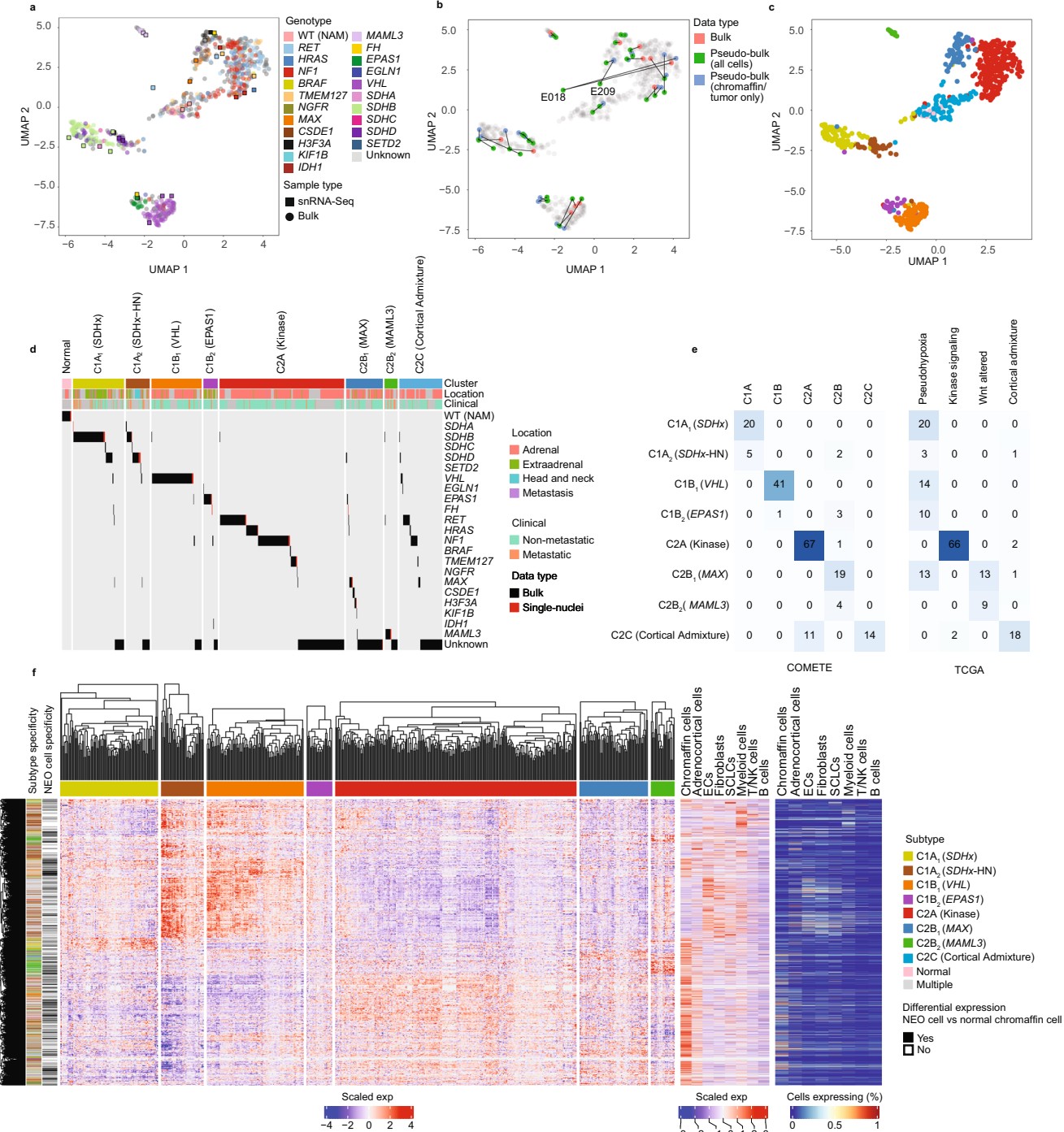

**Fig. 2 | PCPG subtyping by consensus clustering of bulk-tissue and pseudo-bulked snRNA-seq gene-expression data. a** UMAP projection of all bulk-tissue and snRNA-seq pseudo-bulked samples. Each dot represents an individual sample, colored by genotype (nBulk = 735, n-snRNA = 32) (WT (NAM): Wild-type normal adrenal medulla). **b** UMAP highlighting the relationships between paired bulk-tissue and pseudo-bulk snRNA-seq (NEO only and all cells). 14/32 samples analyzed with snRNA-seq had bulk-tissue gene-expression data available. Colors indicate the data type and data points from the same sample are linked by lines. (nBulk = 735, n-snRNA = 32) **c** UMAP illustrating tumor subtypes identified with consensus clustering, colored according to subtype as per panel **d** (nBulk = 735, n-snRNA = 32). **d** PCPG subtypes and gene driver mutations including bulk-tissue and pseudo-bulk snRNA-seq samples. Annotation bars indicate the tumor subtype and anatomical location of the primary tumor. Driver mutations, primary tumor anatomical location and metastatic status were derived from the publicly available metadata. Normal represents normal adrenal tissue. **e** Intersection of PCPG subtypes from this study with previous PCPG subtyping efforts, conducted by the COMETE[10] (left panel) and TCGA (right panel) groups[15]. Cell numbers and color intensity indicate the number of overlapping samples. **f** PCPG subtype-specific DE genes, (identified with a bulk-tissue one subtype versus rest comparison, (absolute log2FC > 0.5, BH adj. *P*-value < 0.05) and intersecting tumor-specific gene expression (determined by pseudo-bulked NEO cells (one subtype) *versus* NAM chromaffin cell comparisons, absolute log2FC > 0.5, BH adj. *P*-value < 0.05). Left panel: Heatmap gene expression (Z-score, standard deviations from the mean) for genes found DE bulk-tissue RNA data (nGenes = 4367, nSamples = 628). Far left annotation bar shows DE genes color-coded by the associated subtype. Black bar indicates if genes were DE in the NEO vs NAM chromaffin cell comparison (in at least one case where a gene is significant in >1 comparison). Right panel: Expression of DE PCPG subtype genes in the non-neoplastic cell types from snRNA-seq data. Plot on left showing z-score scaled expression for pseudo-bulk data and plot on the right showing fraction of cells expressing the gene.

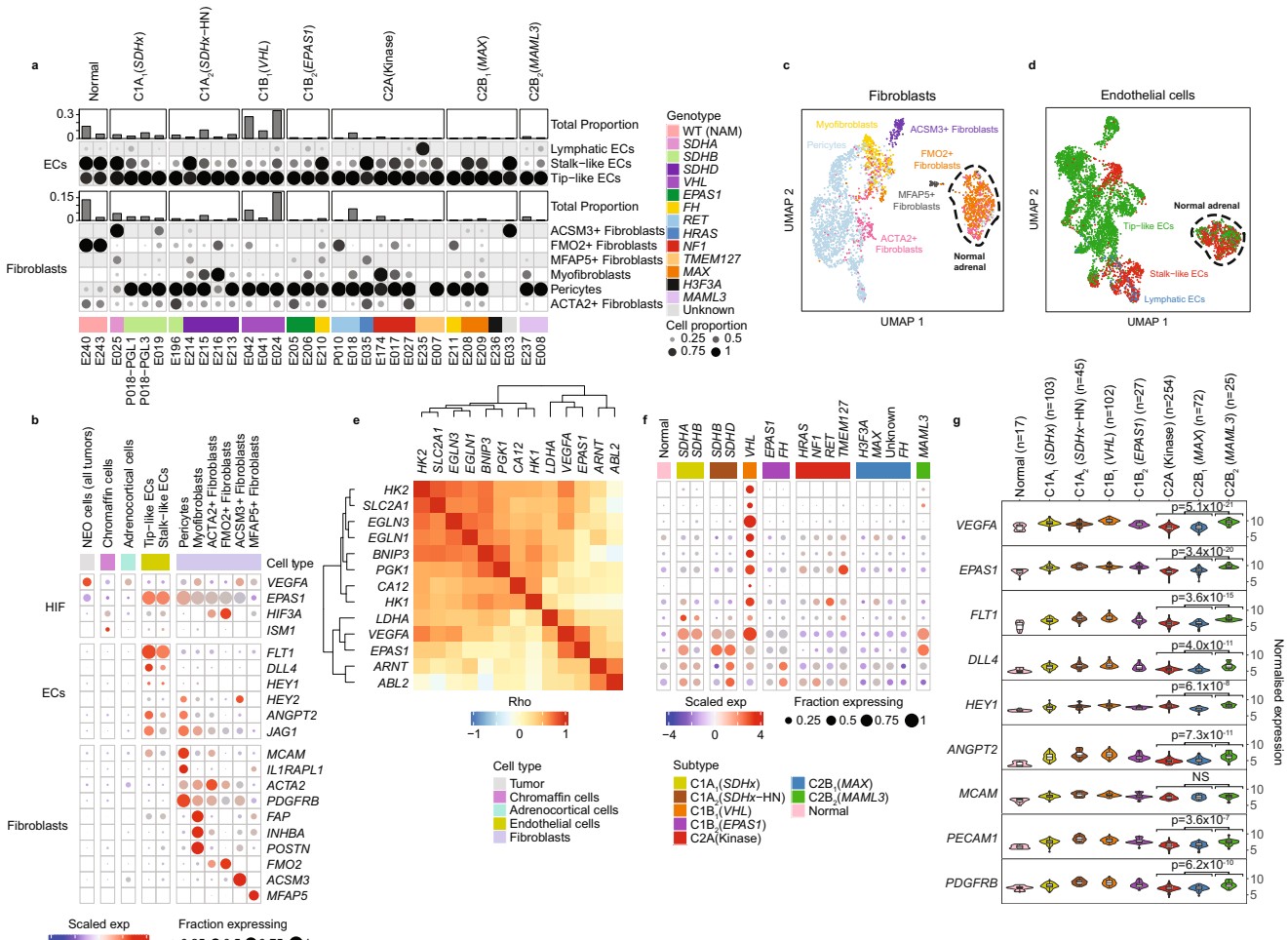

**Fig. 3 | Analysis of EC and fibroblast cells in PCPG and NAM tissues. a** Supervised classification and enumeration of fibroblast and EC cells in PCPG. Proportions of fibroblasts and ECs within each sample (top panel) and the functional subtypes that make up each major cell type (bottom panels). **b** Dot plot of the DE cell type/ pathway marker genes of interest expressed in each functional cell subset. Top ranked DE genes contrasting fibroblasts and ECs from tumor and NAM tissues, respectively, in addition to pro-angiogenesis genes *EPAS1*, *VEGFA* and the anti-angiogenesis genes *ISM1*, *HIF3A* found DE between NEO and normal chromaffin cells from NAM tissues. Dot plot shows expression from nuclei combined from multiple tumors according to cell type as well as NAM tissues. Chromaffin cells shown are from NAM tissues only. **c, d** UMAP reclustering of **c** fibroblasts (nCells = 2901) and **d** ECs (nCells = 5430) from all tumor and normal samples analyzed by snRNA-seq. Nuclei are colored according to cell subtype classification.

**e** Spearman-rank correlation for HIF-related genes in NEO cells. **f** Pseudo-bulk expression across PCPG samples (NEO cells from tumors combined for genotype/ subtype analysis) for the genes described in (**e**). **g** Bulk-tissue gene-expression of angiogenesis gene markers as well as *EPAS1* and *VEGFA* within PCPG subtypes (nSamples = 645) (The lower and upper hinges of each boxplot correspond to the first and third quartiles, respectively, and the median value is marked. The whiskers extend to the largest and smallest value not greater than 1.5 times the interquartile range above or below the upper and lower hinges, respectively. Values beyond the whisker extents are deemed outliers and are plotted individually). WT (NAM): Wild type normal adrenal medulla. Differential gene expression was determined by limma using empirical Bayes moderated *t*-statistics, and adjusted for FDR with Benjamini-Hochberg (BH) correction.

aggressive clinical features[15], but did not reach statistical significance (Fisher exact test BH-adj, P-value <0.05) in our analyis (Supplementary Table 1). Finally, subtype C2C overlapped with the TCGA adrenocortical admixture group[15] and included NAM samples (bulk-tissue and snRNA-seq analyzed). Accordingly, high numbers of adrenocortical cells (29–44% of all cells) were detected in NAM tissues by snRNA-seq. The C2C subtype was ignored from further analysis given the con-founding nature of the adrenocortical cells in these samples and therefore we concluded there are likely seven true PCPG gene-expression subtypes.

Differential expression (DE) analysis was applied to bulk-tissue data using a one subtype versus rest architecture (Log2FC > 0.5, BH-adj. *P*-value < 0.05) (Fig. 2f, Supplementary Data 5). Many subtype DE genes were overexpressed in stromal and immune cells detected by snRNA-seq, indicating enrichment of these cells in PCPG, as discussed below. To identify NEO cell-specific genes, snRNA-seq data was used to

contrast pseudo-bulked NEO cells (combining nuclei from multiple samples for PCPG subtypes) and pseudo-bulked NAM chromaffin cells (log2FC > 0.5, BH-adj. *P*-value < 0.05) (Supplementary Data 6). Between 439 and 2256 genes were DE between NEO cells and NAM chromaffin cells, with 38 genes commonly DE in all PCPG subtype comparisons.

## PCPG tumor microenvironment is dominated by pro-angiogenic cell types

To further dissect the identity and functional state of stromal and immune cell types in PCPG we used supervised classification employing cell type references from two published cancer datasets[30,31] (see Methods). The frequency and proportions of stromal cell subsets in PCPG tumors is shown in Fig. 3a and Supplementary Data 2 and top cell type marker genes in Fig. 3b.

UMAP clustering of fibroblasts and ECs showed distinct tran-scriptional differences in these cells originating from tumors and NAM

tissues (Fig. 3c, d). Myofibroblasts were detected in tumors (0.02–1.49% of all cells) but not NAM tissues. DE genes between tumor and NAM-associated fibroblasts (log2FC > 0.5, BH adj. *P*-value < 0.05) included *FAP*, *INHBA* and *POSTN* and these genes were accordingly expressed in myofibroblasts (Fig. 3b, Supplementary Data 6). Smooth-muscle actin (*ACTA2*) expressing fibroblasts were detected in tumors (0.35–2.44% all cells) and NAM tissues (0.34–2.5% all cells). Some fibroblasts formed discrete UMAP clusters but were not classified to cell types defined within published cancer references. The majority (73–74%) of fibroblasts in NAM tissues expressed *FMO2*, a fibroblast marker in healthy skin[32]. Two other fibroblast populations included *ACSM3*-expressing cells detected in two PCPGs (E025, SDHA/C1A$_1$; E033, unknown driver/C2Bi) and *MFAP5*-expressing cells found at low frequency in tumors and NAM tissues.

Consistent with neovascularization, *FLT1*-expressing ECs were abundant in PCPG and comprised mostly of *DLL4*-expressing tip-like cells (0.049–23.83% all cells). Pericytes, known to be important for vessel formation, expressed *MCAM* and *ILRAPL1* and were the dominant fibroblasts in tumors (0.06–14.32% all cells) (Fig. 3a, b). DE between tumor and NAM ECs (log2FC > 0.5, BH adj. *P*-value < 0.05) also reflected the pro-angiogenic features of PCPG with overexpression of TIE2 receptor ligand *ANGPT2* and coagulation factor *VWF* (Fig. 3b, Supplementary Data 6). Independent analysis of tumor and NAM-derived tip-like or stalk-like EC subsets showed a substantial overlap in DE genes (36.8%) irrespective of the EC subset (Supplementary Data 7, Supplementary Fig. 8), suggesting a common tumor-associated EC program.

Pro-angiogenic factor *VEGFA* was over-expressed in NEO cells and highest in *VHL*-mutant tumors, consistent with a higher number of ECs detected in *VHL*-mutant PCPG compared to other PCPGs (*T*-test FDR < 0.05) (Fig. 3a, b, Supplementary Data 8). *VEGFA* was highly correlated with *EPAS1* expression in NEO cells across all PCPG (Spearman-rank correlation *rho* = 0.72) (Fig. 3e), whereas NEO cells from *VHL*-mutants exhibited expression of additional HIF target genes including *HK1*, *HK2*, *CA12*, *SCL2A1*(GLUT1) (Fig. 3f). Interestingly, angiogenesis inhibitors *ISM1*[33] and *HIF3A*[34] were under-expressed in NEO cells from all PCPG subtypes compared to NAM chromaffin cells (Supplementary Data 6).

Unexpectedly, HIF-related genes *VEGFA* and *EPAS1* were over-expressed in C2B$_2$ (*MAML3*) tumors, while relatively lowly expressed in *EPAS1* and *FH*-mutants, the latter commonly presumed to be pseudo-hypoxic (Fig. 3f, g). Although relatively few ECs and pericytes were detected in C2B$_2$ (*MAML3*) tumors by snRNA-seq, markers of ECs (*FLT1*, *ANGPT2*, *HEY1*, *DLL4*) and pericytes (*MCAM*) were overexpressed in C2B$_2$ (*MAML3*) compared to other C2 subtypes in the bulk-tissue data (Fig. 3g)(BH-adj. *P* < 0.001). C2B$_2$ (*MAML3*) PCPG therefore likely have pro-angiogenic features.

## Immune cell infiltrates in PCPG are predominantly macrophages

With respect to immune cells, myeloid cells were the dominant leukocytes in PCPG (range 0.06–31.05%, mean 7.42%, all cells), consistent with a prior study that showed abundant monocytes in PCPG[35]. Most myeloid cells detected by snRNA-seq were classified as macrophages (94% of total myeloid cells) with minor populations of CD16+ (1.9%) and CD16− (2.5%) monocytes, IDO1+ (1%) and CD1C+ (1.9%) dendritic cells and mast cells (*TPSAB1*)(3%)(Fig. 4a, b, Supplementary Data 2).

Macrophages are known to exhibit significant plasticity within a spectrum of polarized states. Visualizing the canonical macrophage marker genes, including so-called M1 and M2 markers, showed considerable heterogeneity across tumors (Fig. 4c). In bulk-tissue gene-expression data, tumors of the C1A$_2$ (*SDHx*-HN) and C1B$_1$ (*VHL*) subtypes had higher expression of the macrophage markers (Fig. 4d). To further validate our observations by IHC, CD68, CD163, and CD206 staining was applied in 12 matched tumors and two normal adrenal tissues (Fig. 4e, Supplementary Table 2) (see Methods).

CD163+ and CD206+ cells were far more abundant in the adrenal cortex compared to the medulla, which explains a relatively high expression of these M2 macrophage markers in NAM bulk-tissue gene-expression data and NAM tissue macrophages by snRNA-seq (Supplementary Fig. 9). Among PCPG tumors, intratumoral CD163+ and CD206+ cells were highest in *VHL*-mutant PCPG (Fig. 4e) compared to other subtypes.

Genes overexpressed in tumor-associated macrophages compared to normal adrenal macrophages included *PLXDC1* and *PLAU* (log2FC > 0.5, BH adj. *P*-value < 0.05) (Supplementary Data 7, Fig. 4c). PLXDC1 is a transmembrane receptor for the pluripotent factor PEDC that has important anti-angiogenic and anti-tumor functions[36,37]. PLAU is a serine protease important for tissue remodeling and angiogenesis[38]. Like our observation of canonical macrophage gene expression across subtypes in the bulk gene-expression data, *PLXDC1* and *PLAU* were highest expressed in C1A$_2$ (HN-PGL) and C1B$_1$ (*VHL*) subtypes (Fig. 4d).

Lymphocytes represented a much smaller fraction of immune cells in tumors by snRNA-seq (range 0.02–8.93%, mean 2.3%). CD3 IHC staining in 12 PCPG confirmed the relatively low number of T cells (mean = 5 cells/mm$^2$) compared to CD163+ macrophages (mean = 105 cells/mm$^2$) (Fig. 4f, Supplementary Table 2). Phenotyping of T cells by snRNA-seq showed predominant *CD4* expressing cells with minor populations of cytotoxic T cells (*CD8A*, *GZMB*), NK cells (*NCAM1*(CD56), *GNLY*) and T-regulatory cells (*FOXP3*) detected (Fig. 4a, Supplementary Fig. 10, Supplementary Data 2). B cells formed three distinct clusters: follicular (*MS4A1*), GZMB+ and plasma cells (*FCRL5*) (Fig. 4e). Interrogation of lymphocyte markers in bulk-tissue data did not show appreciable differences across PCPG subtypes but the innate cell marker *GLNY* and cytolytic marker *GZMB* were slightly elevated in C2B$_2$ (*MAML3*) and C1A$_2$ (*SDHx*-HN) tumors as well as occasional outlier cases in other PCPG subtypes (Fig. 4f).

## Schwann-cell-like cells (SCLCs) and putative paracrine signaling with NEO cells

PCPG, neuroblastoma, and NAM tissues contain Schwann-like cells called sustentacular cells detected by IHC staining for SOX10 and S100[39,40]. In mice and humans, chromaffin cells are thought to arise from pluripotent neural-crest cells called Schwann-cell precursors (SCPs) that also express these markers[41–43]. Whether sustentacular cells in PCPG are precursors or terminally differentiated cannot be confirmed in our data, therefore we described them as Schwann-cell-like rather than SCPs.

SCLCs were abundant in some C1 PCPG (range 0–22.9%, mean 6.8%) but infrequent in C2 PCPG (range 0–3.89% mean 0.85%), although not statistically significant between subtypes (Fig. 5a). InferCNV analysis of the snRNA-seq data showed that SCLCs were ostensibly diploid in PCPG (Fig. 5b), consistent with prior observations using orthogonal approaches such as IHC[44] and flow cytometry[45]. *CDH19* was expressed by SCLCs and RNA in situ hybridization (ISH) targeting *CDH19* in an AT-PG tumor (P018-PGL3) showed staining of spindle-like cells, that were similar in morphology to S100 + cells using IHC (Supplementary Fig. 11). *CDH19* and *SOX10* expression in the bulk-tissue data supported a trend towards a higher number of SCLCs being present in C1 PCPG subtypes. C1B$_2$ (*EPAS1*) PCPG had fewer SCLCs by snRNA-seq and low *SOX10*/*CDH19* expression in the bulk-tissue data (Fig. 5c).

To investigate cell–cell communication in PCPG, we inferred cell–cell interactions using the NATMI method[46], ranking receptor-ligand gene pairs based on the mean edge total expression weight for receptor-ligand interactions (Supplementary Data 9). Inferred receptor-ligand interactions between normal chromaffin cells and SCLCs in NAM tissues was 8% of total edges (Fig. 5d) and 12% of edges between NEO cells and SCLCs in PCPG (Fig. 5e). Receptors and ligands significantly overexpressed in SCLCs (log2FC > 3, BH adj. *P*-value <

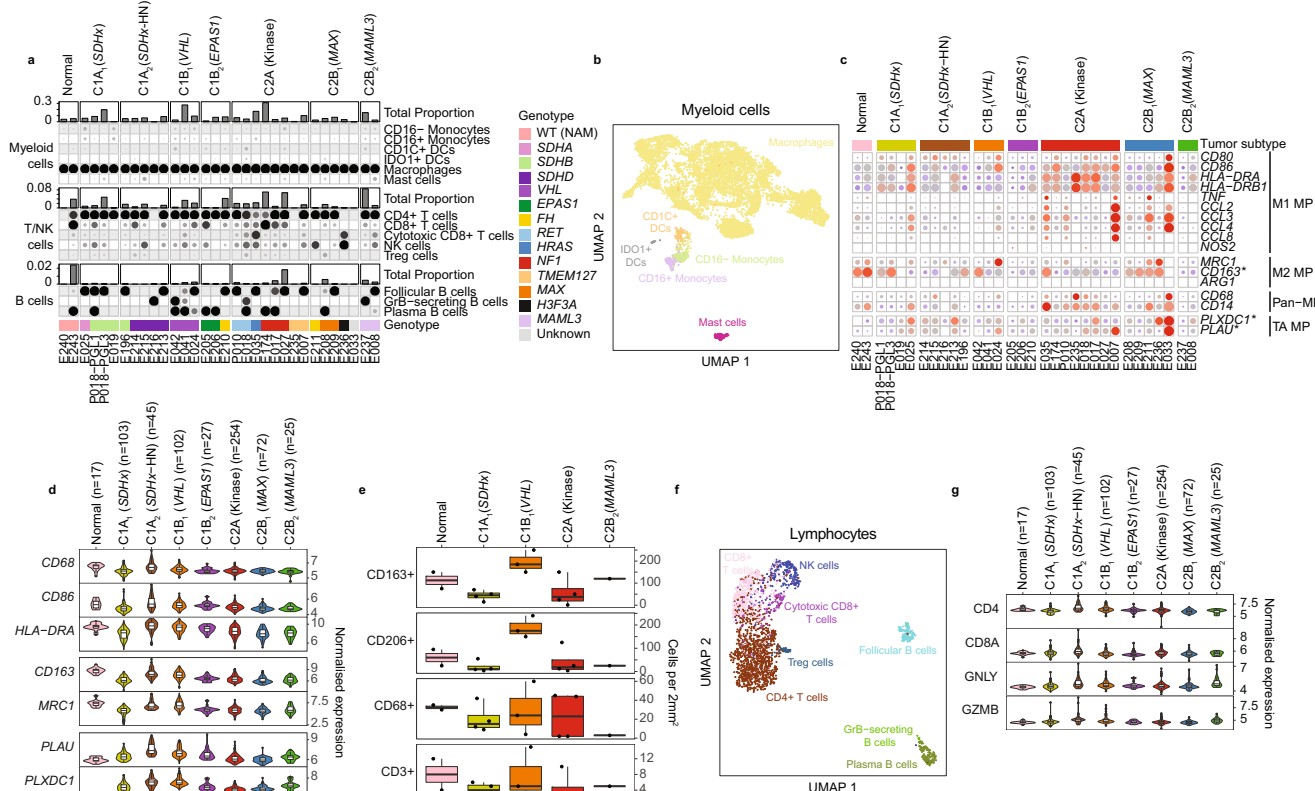

**Fig. 4 | Analysis of leukocytes isolated from PCPG and NAM tissues by snRNA-seq. a** Supervised classification of immune cell fractions in PCPG. Proportions of each major immune cell types within each sample (top panel) and the functional subsets that make up each major cell type (bottom panels). **b** UMAP re-clustering of myeloid cells from all snRNA-seq samples. Nuclei are colored according to their classified functional cell subsets (nCells = 7977). **c** snRNA-seq expression of macrophage (MP) marker genes in macrophages detected on PCPG and NAM tissues. (TA: Tumor-associated) **d** Enumeration of intra-tumoral leucocytes detected by immunohistochemistry staining for canonical macrophage markers (CD163, CD206, CD68) and a T cell marker (CD3) (nSamples = 645, the lower and upper hinges of each boxplot correspond to the first and third quartiles, respectively, and the median value is marked. The whiskers extend to the largest and smallest value not greater than 1.5 times the interquartile range above or below the upper and lower hinges, respectively. Values beyond the whisker extents are deemed outliers and are plotted individually). **e** Bulk-tissue expression of DE expressed macrophage genes contrasting tumor and NAM macrophages (nSamples = 14, box plots to be interpreted as per panel **d**). **f** UMAP reclustering of T/B cells from all snRNA-seq samples (nCells = 2140). **g** Bulk-tissue gene-expression of T/NK -cell markers across subtypes. DCs dendritic cells, NK cells natural killer cells. WT (NAM) wild type normal adrenal medulla (nSamples = 645, box plots to be interpreted as per panel **d**).

0.05) were intersected with NATMI results to identify the top gene-gene pairs between SCLCs and NEO cells (Fig. 5f). Ligand-receptor pairs reflected known Schwann-cell function including *LGI4-ADAM22/ ADAM23*, *GDNF-RET*[47], and *FGF2-FGFR1/FGFR2*[48]. WNT (*RSPO3-LGR4/ LGR5*) and TGFB signaling (*TGFB2-TGFBR1*) also featured. Examples of receptors overexpressed in SCLCs included *ERBB3* (tumor ligands *NRG1*, *NRG2*) and GDNF receptor 1 encoding *GFRA1* (ligand *NCAM1*).

**PCPG exhibit variable chromaffin cell differentiation patterns**

PCPG variably express chromaffin-related genes indicating divergent states of cellular differentiation or developmental origins. To determine the similarity of PCPG to sympathoadrenal cells during early development we used a published snRNA-seq dataset of normal human fetal adrenal tissues taken at seven developmental time points[49]. Jansky et al. identified adrenal medullary cell types including SCPs (cycling, late), bridge cells, connecting progenitor cells, chromaffin cells (early and late), and neuroblast populations (early, cycling, and late) (Fig. 6a).

We used two approaches to compare NEO cells to normal fetal adrenal cells. Firstly, we applied a supervised cell classification method like that used for classifying stromal and immune cells (Fig. 6b, c, Supplementary Data 10). As expected, nearly all (99%) adult NAM chromaffin cells classified as late chromaffin cells and 95% of SCLCs classified as SCPs (89% late SCPs). Most NEO cells from C1 PCPG were

classified as early chromaffin cells (range 38–92%, mean 66%). In contrast, NEO cells from C2A (Kinase) and C2B₁ (*MAX*) tumors were mostly classified as late chromaffin cells, except for two *NF1*-mutant PCs (E027, E174) and a PC of no known gene driver (E033) that were mostly comprised of NEO cells classified as early chromaffin cells. The C2B₂ (*MAML3*) PC had a higher fraction of early chromaffin cells (mean 56%), whilst neuroblasts (range 5–17%) and connecting progenitor cells (range 1–9%) were minor subsets. *FH*-mutants were divergent with one case (E210) composed mostly of early chromaffin cells (62%) and the other (E211) predominantly late chromaffin cells (92%).

A second approach involved calculating module-scores for cell type marker gene-sets (see Methods). Cell type module-scores in NEO cells were calculated using gene-sets identified by Jansky et al. and data for NEO cells was plotted pooling cells by PCPG subtype (Fig. 6d, Supplementary Data 11). The same gene-sets were also used for gene-set variance analysis (GSVA) calculating enrichment scores in individual samples within the bulk-tissue data (Fig. 6e). The expression of select sympathoadrenal cell type markers was visualized in our snRNA-seq data (NEO cells only) and the bulk-tissue data, respectively (Fig. 6f, g).

Enrichment of early chromaffin genes was observed in C1 subtypes, consistent with the cell classification approach. C1B₁ (*VHL*) tumors also had enrichment of the connecting progenitor cell gene-set; representing a transient cell population connecting bridge,

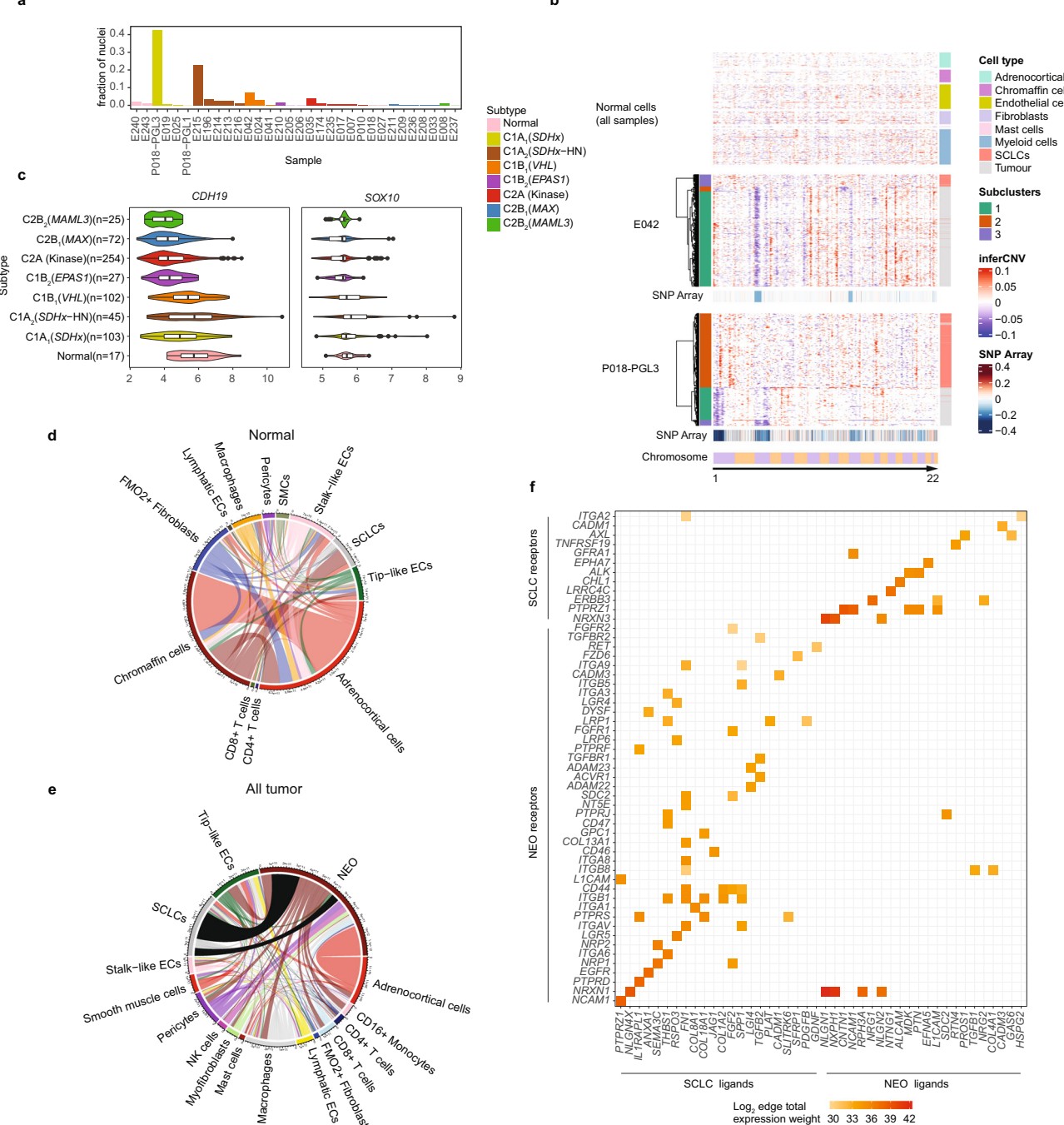

**Fig. 5 | SCLCs and cell-signaling in PCPG and NAM tissues. a** Fraction of SCLCs among all cells detected in PCPG tumors by snRNA-seq. **b** Inferred copy-number and clustering of NEO and SCLC populations in two PCPG tumors (E042, P018-PGL3) (bottom panels; clustered tumor cells and SCLCs, top panel; non-neoplastic cell types excluding SCLCs from all tumors and normal tissues). InferCNV scale = modified expression, SNP array scale = log ratio. **c** Gene-expression of SCLC marker genes *CDH19* and *SOX10* in bulk-tissue data (nSamples = 645, the lower and upper hinges of each boxplot correspond to the first and third quartiles, respectively, and the median value is marked. The whiskers extend to the largest and smallest value

not greater than 1.5 times the interquartile range above or below the upper and lower hinges, respectively. Values beyond the whisker extents are deemed outliers and are plotted individually). Chord plots of NATMI mean edge total expression weights showing the average of the total amount of inferred signaling between all cell types in **d**, NAM samples and **e**, PCPG tumor samples. **f** Mean edge total expression weight of ligands and receptors expressed by SCLCs and their known target receptors and ligands on tumor cells. SCLCs ligands and receptors were DE in SCLCs vs the rest of normal cells at log2FC > 3, BH adj. *P*-value < 0.05).

chromaffin and neuroblast cell types detected at 7–8 weeks post-conception[49]. As expected, C2A (Kinase) tumors had higher module scores for late chromaffin cells and expressed *PNMT*, encoding the enzyme that converts norepinephrine to epinephrine, as well as the neuroblast markers *RET* and *NPY*. Notably, *NPY* expression was lower in PCPG C1 subtypes but retained in C1B₂ (*EPAS1*) tumors. C2B₂ (*MAML3*)

PCPG had low expression of the chromaffin cell markers *CARTPT* and *DLK1* but high expression of early neuroblast markers including *ALK*, *RET*, and *NTRK3*. A subset of PCPG had expression of bridge cell markers (*ASCL1*, *CDH9*, *ERBB4*), including tumors of the C2B₁ subtype, for example, a tumor with an *H3F3A* mutation (E326) and another without a known gene driver (E033)[50].

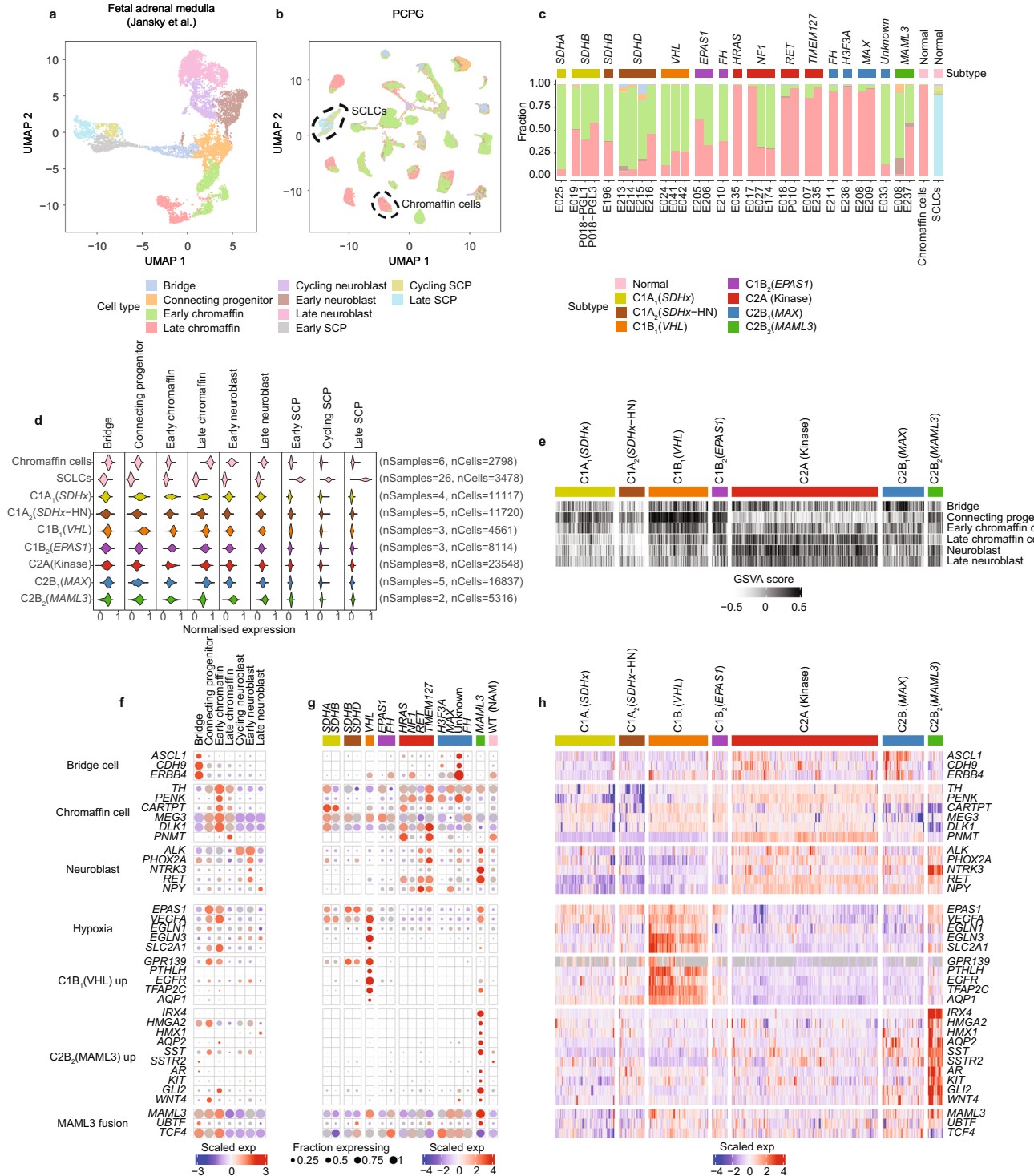

**Fig. 6 | Comparison of expression profile of PCPG neoplastic cells to normal fetal adrenal cell types. a**, UMAP projection illustrating snRNA-seq of fetal adrenal medulla (nSamples = 17, nCells = 10,739), previously published by Jansky et al.[49]. **b** UMAP of PCPG cell populations colored by their cell type classification (nSamples = 30, nNeoplastic cells = 81,213, nSCLCs = 3335) and adult adrenal medulla (nSamples = 2, nChromaffin cells = 2798, nSCLCs = 143). The black dotted lines show normal chromaffin and SCLC cell types. **c** Proportion NEO cells or normal cells (PCPG and NAM combined) classified as normal fetal adrenal cell types. **d** snRNA-seq gene-module scores for NEO cells, normal chromaffin cells, and SCLCs based on the gene-sets identified by Jansky et al. (nSamples = 32, nCells = 87,489)[49]. **e** GSVA scores using fetal adrenal gene-sets across PCPG subtypes in the PCPG bulk-tissue data. **f** snRNA-seq gene-expression for markers for fetal adrenal cells and PCPG subtypes in normal fetal cell data. **g** Expression of the same genes shown in panel (**f**) but in NEO cells across PCPG subtypes (NEO cells were pooled based on genotype and subtype profile). **h** Bulk-tissue gene-expression (*n* = 628 samples) for the same genes shown in panel (**f**, **g**).

Consistent with HIF-2α being important for development of sympathetic and parasympathetic tissues[51,52] *EPAS1* and *VEGFA* were overexpressed in connecting bridge progenitors and early chromaffin cells (Fig. 6f, g). Remarkably, many other genes DE between PCPG subtypes were either undetected or very lowly expressed in developing sympathoadrenal cell types compared to NEO cells. For instance, C1B₁(VHL) DE genes including *PTHLH, TFAP2C,* and *AQP1* were lowly expressed in both fetal and adult chromaffin cells. As previously

described, C2B₂ (*MAML3*) PCPG over-expressed Wnt and Hedgehog pathway genes including *GLI2* and *WNT4*[15] and these were also lowly expressed in normal cell types. C2B₂ (*MAML3*) tumors uniquely over-expressed several transcription factors including *HMX1*, *HMGA2*, *IRX4*[25] as well the water channel protein *AQP2*, the somatostatin ligand (*SST*) ligand and its cognate receptor *SSTR2*, the latter being very lowly expressed in C1B₁ (*VHL*) tumors. Notably, *MAML3* and the fusion-partner genes *UBTF* and *TCF4* were expressed in bridge, connecting progenitor and early chromaffin cell types.

## Cell receptors identified as putative theranostic targets in PCPG tumors

Numerous genes overexpressed in NEO cells encode cell surface receptors; an attractive class of therapeutic or diagnostic (theranostic) targets (Fig. 7a, Supplementary Data 6). Among tyrosine kinases, *EGFR* was overexpressed by C1A₂ (*VHL*) tumors and some C1A₁ (*SDHx*) tumors, while *KIT* was highly expressed in the C2B₂ (*MAML3*) subtype. *EGFR* overexpression has also been described in *VHL*-associated renal cell carcinomas[53]. The androgen receptor (*AR*) and G-protein coupled receptor *VIPR2* were also highly expressed in C2B₂ (*MAML3*) tumors. The orphan G-protein coupled receptor *GPR139* had restricted expression to C1 PCPG. In the developing fetal adrenal snRNA-seq data, *GPR139* expression was expressed in connecting progenitor and early chromaffin cells but absent in late chromaffin cells (Fig. 6f). Comparative analysis across TCGA pan-cancer gene-expression data confirmed *GPR139* was highly expressed in subsets of PCPG, while also in CNS malignancies and a subset of breast adenocarcinomas (Fig. 7b).

## Transcriptional patterns associated with metastatic PCPG

Clinical annotation available for the bulk-tissue gene-expression data enabled DE analysis between metastatic (*n* = 52) and non-metastatic PCPG (*n* = 330). Given the significant transcriptional heterogeneity across PCPG subtypes and the higher rate of metastatic disease in C1A (*SDHx*) subtypes, C1A (*SDHx*) tumors were analyzed independently of

non-*SDHx* tumors. Interestingly, a total of 299 genes were DE between metastatic and non-metastatic C1A (*SDHx*) tumors (Log2FC > 0.5, BH-adj. *P*-value < 0.05) (Fig. 8a), whereas only 47 genes were DE in the non-*SDHx* analysis and only three genes overlapping in both analyses (Supplementary Fig. 13, Supplementary Data 12). GSVA was also done using MSigDb Hallmark gene-sets as well as stromal, immune, and fetal adrenal cell gene-sets[49] (Fig. 8b, Supplementary Table 3). Among the non-*SDHx* cases no gene-sets were significant between metastatic and non-metastatic tumors. Metastatic C1A (*SDHx*) tumors had elevated cell cycle-related gene-sets (Hallmark G2M, E2F targets, Mitotic spindle; Jansky cycling neuroblast) (Log2Fold > 1.0, BH-adjusted *P*-value < 0.05) and overexpressed canonical proliferation markers (*MKI67*, *TOP2A*) (Fig. 8a). Conversely, SCLC and SCP gene-sets and Schwann-cell marker genes (*CDH19*, *SOX10*) were down in metastatic C1A (*SDHx*) group (Fig. 8a, b). Other genes overexpressed in metastatic C1A (*SDHx*) tumors included collagens (*COL1A1*, *COL6A3*); overexpressed in fibroblasts and occasionally NEO cells in our snRNA-seq data (Supplementary Fig. 12, Supplementary Data 12), metalloproteases (*MMP9*, *MMP12*) expressed at low levels in all cells in the snRNA-seq data, the EMT transcription factor *TWIST1* as well as the polycomb repressor *EZH2*, variably expressed in NEO and stromal cell types (Supplementary Fig. 12). Macrophage markers *MARCO* and *CD68* were overexpressed in the metastatic C1A (*SDHx*) tumors. Interestingly, the cell surface receptor *GPR139* was also overexpressed in metastatic C1A (*SDHx*) tumors.

## Discussion

In this study, we confirmed a strong association between PCPG driver genes and transcriptional programs within neoplastic and non-neoplastic compartments. Importantly, our analysis increased the resolution of PCPG subtyping, identifying distinct PCPG clusters associated with rare PCPG driver genes, including *EPAS1* (C1B₂), *FH* (C1B₂, C2B₁), *MAML3*-fusions (C2B₂) as well as parasympathetic *SDHx* HN-PG(C1A₂)[9,10]. Angiogenesis driven by HIF-pathway dysregulation is a dominant feature in PCPG and we have described these features at single-cell resolution. We confirmed PCPG neoplastic cells resembled

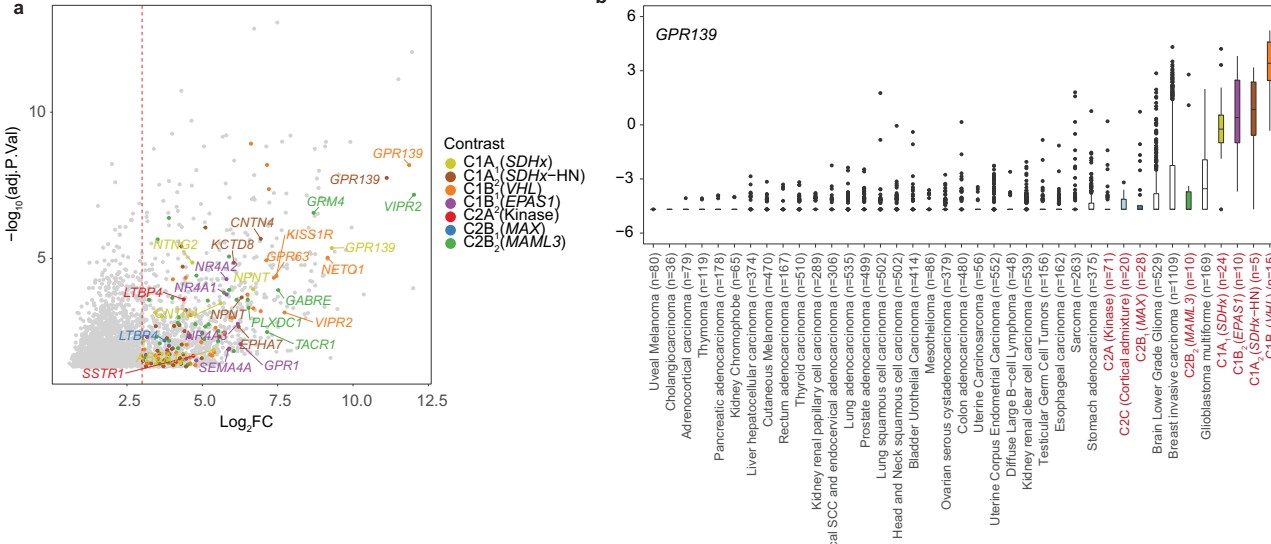

**Fig. 7 | Identification cell surface receptors as putative therapeutic targets.**
**a** Volcano-plot showing significantly DE genes (adj. *p*-value < 0.05) found by contrasting pseudo-bulked NEO cells and NAM chromaffin cells based on PCPG subtype (Supplementary Data 5). Only those genes encoding cell surface receptors are colored, with top receptors in each PCPG subtype labeled. The dotted line shows log2 threefold change threshold used for selecting receptor genes. **b** Expression of *GPR139* across tumor types in the TCGA pan-cancer dataset of 10,211 tumors from

32 cancer types. PCPG are colored by their respective subtype. WT (NAM): Wild type normal adrenal medulla (The lower and upper hinges of each boxplot correspond to the first and third quartiles, respectively, and the median value is marked. The whiskers extend to the largest and smallest value not greater than 1.5 times the interquartile range above or below the upper and lower hinges, respectively. Values beyond the whisker extents are deemed outliers and are plotted individually).

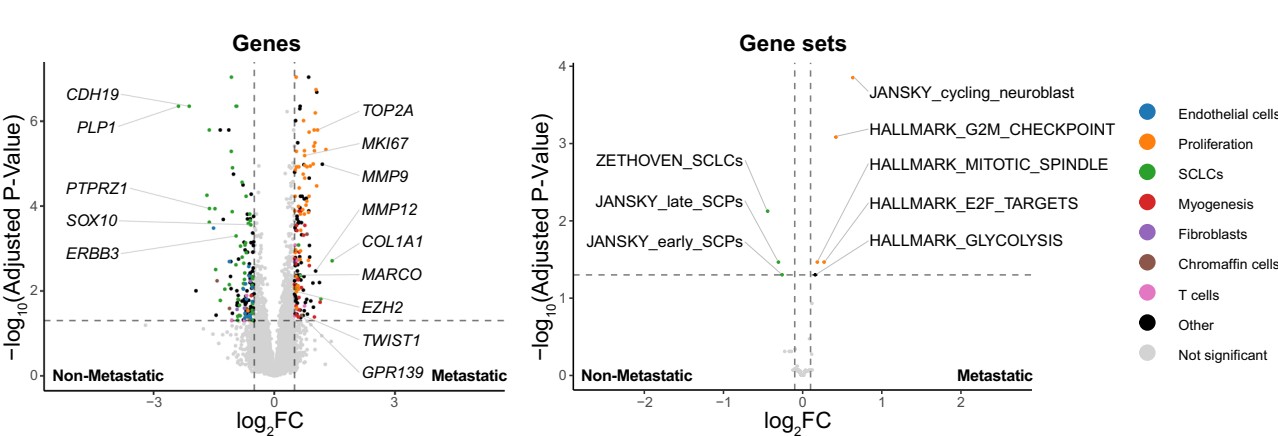

**Fig. 8 | Differentially expressed genes and gene-set scores distinguishing non-metastatic and metastatic C1A (*SDHx*) PCPG in bulk tissue analysis. a** Volcano-plot depicting DE genes between non-metastatic and metastatic PCPG (BH-adj. *P*-value < 0.05, log₂FC > 0.5). **b** Differential GSVA scores for gene-sets observed contrasting non-metastatic and metastatic PCPG (BH-adjusted *P*-value < 0.05, log₂FC > 0.1) in only C1A (*SDHx*) samples. In panels **a** and **b** gene symbols and pathways are color coded based on biological processes or cell type association.

normal chromaffin cells; however, they can have heterogeneous transcriptional profiles expressing markers of early chromaffin and neuroblast cells as well as genes that are not expressed in adult or fetal chromaffin cells. Furthermore, by using a large series of bulk-tissue gene-expression data we have explored differential expression in metastatic and non-metastatic PCPG, identifying diagnostic or therapeutic biomarkers that provide important leads for further investigation.

Consistent with HIF-pathway activation, tumoral *VEGFA* expression, and abundant vascular cell types were observed in PCPG. *VHL*-mutant PCPG exhibit the strongest induction of HIF target genes, consistent with the canonical HIF-regulatory function of VHL. Together with loss of the *VHL* locus in neoplastic cells (Supplementary Fig. 4), these findings support its classic tumor-suppressor function in PCPG, which has been previously questioned owing to the unique genotype-to-phenotype associations in PCPG compared to other VHL syndromic tumors[54]. Uniquely, we found *VEGFA* and *EPAS1* overexpressed in *MAML3* fusion-positive PCPG, which challenges the commonly held view that HIF-pathway activation is restricted to C1 PCPG. The mechanism for HIF-pathway activation in *MAML3* fusion-positive PCPG is unclear. MAML3 is a transcriptional co-activator of NOTCH signaling and although NOTCH signaling can induce *EPAS1* expression[55], loss of *MAML3* exon 1 encoding the N-terminal NOTCH domain is a recurrent in *MAML3*-fusions, making NOTCH induction of *EPAS1* seem unlikely. *MAML3*-fusion functionality may also be dependent on the 5′-fusion partner gene including *UBTF*, a nucleolar transcription factor involved in RNA polymerase 1 ribogenesis, and transcription factor *TCF4*[15]. Notably, as *EPAS1* and *VEGFA* are expressed during early development of adrenal and extra-adrenal paraganglia[51,52] the transcriptional profile of NEO cells may reflect, at least in part, an early developmental program rather than *MAML3*-fusion induced HIF-pathway activation. Clearly further experimentation will be required to determine a potential link between *MAML3*-fusions and the HIF-pathway in PCPG.

Reliable clinical biomarkers of metastatic progression are still needed in PCPG. With respect to PCPG subtypes, the C1A (*SDHx*) and C2B₂ (*MAML3*) subtypes have a higher propensity to develop metastatic disease[15]. Morphological features, IHC stains, gene-expression, somatic gene mutations (principally *TERT* promoter and *ATRX* mutations) as well other features have been proposed for risk stratification[7]. Our observation of cell-cycle and proliferation-related markers in metastatic PCPG is wholly consistent with the reported association between elevated Ki67 staining and increased risk of metastasis[56].

Similarly, a reduced number of sustentacular cells reported in metastatic PCPG[57] is concordant with lower SCLC marker gene expression observed in bulk-tissue gene-expressioin data. Other genes overexpressed in metastatic PCPG included genes associated with tissue remodeling and EMT, also consistent with a previous study[58]. Very low T cell infiltrates in most PCPG may predict limited benefit from immune checkpoint inhibitors, consistent with the modest responses to these drugs observed in PCPG patients to date[59,60]. Macrophages were abundant in PCPG but are highly heterogenous in their expression profile, with macrophage marker expression highest in PCPG tumors with neoangiogenic features, although not ubiquitous across all pseudohypoxic subtypes, including a significant fraction of the C1A (*SDHx*) group. Further immunohistochemical staining in a larger series is required to deconvolve the relative spatial context of macrophage infiltrates within the PCPG and potential associations with the metastatic phenotype.

Importantly, we identified promising biomarkers that may inform future treatment strategies in metastatic PCPG. Tyrosine kinase and HIF-2α inhibitors have been proposed for treatment of pseudohypoxic PCPG[61], therefore, a similar rationale may be extended to C2B₂ (*MAML3*) tumors, which have a higher propensity to develop metastatic disease. G-protein coupled receptors are an attractive class of therapeutic targets and among this group *GPR139* expression is quite novel. GPR139 synthetic agonists and antagonists have already been identified, therefore direct pharmacological intervention is plausible[62]. Alternatively, radionuclide-labeled small molecules, peptides or antibodies targeting GPR139 or other identified cell surface receptors may be used, analogous to targeting SSTR2 using ⁶⁸Ga-DOTATATE in PCPG[63]. Finally, because development of treatment strategies often begins with histopathological assessment of biomarker protein expression, our findings may expand the list of emerging biomarkers applicable by pathologists in the diagnostic workup of PCPG.

## Methods
### Patient samples
Research was done under a protocol approved by the human research ethics committee at Peter MacCallum Cancer Centre and under the guidelines of the National Health and Medical Research Council in accordance with the Helsinki Declaration of 1975, as revised in 1983. All patients provided written informed consent for the use of their deidentified biospecimens for research purposes. No compensation was provided for provision of samples. Patient samples were collected

under protocols approved by the respective institutional review boards (IRB). Organizations contributing patient samples included the Victorian Cancer Biobank under protocols approved at Austin Health, Melbourne Health, and Monash Health ($n = 4$), the Peter MacCallum Cancer Centre ($n = 4$), Kolling Institute Neuroendocrine Tumor Bank under a protocol approved at North Sydney Local Health District ($n = 8$), National Institute of Health ($n = 10$), University of Colorado ($n = 1$), University of Texas Health Science Center at San Antonio ($n = 2$), Tufts Medical Centre ($n = 1$), and Palacky University ($n = 2$). (see Supplementary Data 1 for patient and sample information).

### Single-nuclei (sn)RNA-seq

snRNA-seq was performed using the 'Frankenstein' protocol (dx.doi.org/10.17504/protocols.io.bqxymxpw)[64]. Briefly, nuclei from frozen tissues were subject to fluorescence-activated nuclei sorting (FANS) by 4′,6-diamidino-2-phenylindole (DAPI) on a BD FACSaria 2 instrument, sorting between 3000 and 10,000 nuclei per sample. Both diploid and tetraploid populations were selected to account for genome duplication in neoplastic PCPG cells[15]. FAN-sorted nuclei were immediately processed using the 10x Chromium Single Cell 5′ Library & Gel Bead Kit (PN-1000002 following the manufacturer's recommendations (10x Genomics, USA). Once processed, snRNA-seq libraries were sequenced in multiple batches on the Illumina Nova-Seq 6000 (Illumina, USA) using 150 bp paired-end sequencing. Between 895 and 4822 cells were sequenced per tumor achieving near saturation coverage at ~5800 unique sequence reads per cell. scRNA-seq binary base calls (BCL) files were demultiplexed and converted into FASTQ files using BCLtoFastq.

### snRNA-seq primary data analysis

FASTQ sequence data were aligned to a custom hg19 (GRCh37, Cell-Ranger reference genome version 3.0.0, build GRCh37.p13) "pre-mRNA" reference, to account for reads that map to both exonic (mRNA) and intronic (unspliced pre-mRNA) regions. This custom reference was created as described here: https://support.10xgenomics.com/single-cell-gene-expression/software/pipelines/3.0/advanced/references. Alignment and UMI counting were performed using cellranger count (v3.0.2).

snRNA-seq data was subject to quality control and data filtering (see Supplementary Fig. 1 for quality control metrics and thresholds). To detect barcodes that likely contained RNA from two or more cells (i.e. doublets), raw counts for each sample were annotated with 'doublet scores' using Scrublet (version 0.2.1)[65]. Scores were normalized within each sample to the median absolute deviation (MAD) of the raw scores. Potential doublet barcodes were removed from further analysis with Scrublet MAD values <2.

Further quality control filtering was done in the context of major cell type types (determined by UMAP clustering) to account for the range of transcriptional activity within major cell lineages. Filtered cell expression matrices from Cell Ranger for each sample were merged into a single matrix and processed using the Seurat R package (version 3.2.3)[66]. Within each sample, cells were filtered out if mitochondrial genes exceeded a median absolute deviation (MAD) value of 5. Log-scaled gene counts and total counts were normalized to MAD values and the filtering threshold was selected per-barcode based on the raw annotation of scMatch as it was observed that certain immune cell types had significantly lower total RNA counts in the snRNA-seq data sets. A threshold of −4 MAD score for B cells, T cells, mast cells or NK cells, otherwise a threshold of −2.5 was used for all other cell types.

The filtered cell expression matrix was then normalized using the SCTransform method with the mitochondrial gene count percentage as a non-regularized latent variable in the variance stabilizing transformation (VST) model[67]. Cells were then annotated with cell cycle phase scores using Seurat's CellCycleScoring function to provide cell cycle classifications after the effect of total UMI counts per-cell had been reduced in the data. SCTransform was then repeated, with the G2M and S phase scores included as additional non-regularized latent variables.

### Uniform manifold approximation and projection (UMAP) clustering

Variable genes were selected based on a residual variance threshold of 1.3. Principal Component Analysis (PCA) was then performed on the resulting scaled expression values. A shared-nearest-neighbor (SNN) graph was constructed using the Seurat FindNeighbors function with the "Annoy" method, using the first 20 principal components (PCs), a cosine distance metric, and number of nearest neighbors counted of 20[68]. Clusters were identified from the SNN graph using the Seurat FindClusters function with the Louvain algorithm and resolution parameter set to 0.8. The UMAP values were calculated from the top 20 PCs using the uwot R package (version 0.1.8) with cosine distance metric and n.neighbors set to 20. For the cell-type-specific UMAP values, PCA was repeated within each subset and the top 30 PCs were used instead with the same parameters otherwise.

### Harmony batch correction

Raw read counts were normalized with the Seurat R package using the log-normalization method with default parameters. The top 3000 most variable features were then selected using the variance stabilizing transformation method. The resulting subset was scaled and centered, and principal component analysis (PCA) was performed using the default parameters. To ameliorate sample or patient-specific batch effects, the Harmony R package (v0.1.0) was applied to Seurat object using patient identifier as the grouping variable. The resulting Harmony embedding was used to perform UMAP dimensional reduction, neighbor finding, and cluster finding with the first 20 dimensions and resolution of 0.5.

### Stromal and immune cell type classification

Each cell barcode was annotated by cell type based on raw counts using scMatch initially using the FANTOM5 reference data set collapsing to one cell type per UMAP cluster by taking the most common cell annotation in each cluster[69]. Cell type labels (prior to the finer cell subtype analysis) were also refined/curated based on gene markers of known cell types accounting for the potential absence of cell types in the FANTOM5 data set (e.g., chromaffin cells and SCLCs were not represented). Major stromal and immune cell types were then later reclassified using two cancer-related scRNA-seq reference datasets[30,31] (GEO accession IDs GSE131907 and GSE146771). Log transcripts-per-million from GSE146771 and counts from GSE131907 were normalized with the LogNormalize function from Seurat with default parameters collapsed to gene expression centroids by taking the mean value per cell type. In a similar approach to scMatch, cells were scored against each centroid using Spearman correlation on a subset of highly variable genes. Variable genes were selected separately for the immune cell types and for all other non-immune normal cell types by ranking their residual variance within the respective groups within the final SCTransform VST model and selecting the top 3000 genes. Centroids from GSE131907 were used to annotate fibroblast, endothelial cell and B cell subtypes. Centroids from GSE146771 were used to annotate T cell, NK cell and myeloid cell subtypes. Unique clusters that did not correspond well with any of the reference cell types but could not otherwise be identified were manually labeled according to marker genes of those clusters.

### Inference of copy-number variation from snRNA-seq

The inferCNV R package (version 1.2.1) (https://github.com/broadinstitute/inferCNV) was used to estimate cell-specific copy number profiles based on gene expression. Adrenocortical cells, chromaffin cells, endothelial cells, fibroblasts, and myeloid cells from

all samples were used as the reference cell types for inferCNV. SCLCs and tumor cells were processed independently to compare to the reference diploid normal cell types. To compare the inferCNV with other copy number methods, matched Affymetrix Cytoscan HD microarray data was available for 12 samples[26,50]. Raw CEL file data (GEO Accession ID: GSE61594, GSE94378) were processed using the rawcopy R package workflow[70].

**Creation of the bulk-tissue RNA gene-expression compendium**
**Microarray data.** Raw microarray data were obtained from seven GEO accessions (GSE2841, GSE19422, GSE19987, GSE39716, GSE50442, GSE51081, and GSE67066) and one ArrayExpress accession (E-MTAB-733). Affymetrix arrays were read into R (version 4.0.4) with the ReadAffy function from the affy R package (version 1.62.0) then normalized using robust multi-array average (RMA)[71,72]. Agilent two-color arrays were read into R using the read.maimages function from the limma R package (version 3.42.0)[73]. Expression values were then normalized using the backgroundCorrect (with method = 'normexp', offset = 5), normalizeWithinArrays (with method = 'loess'), normalizeBetweenArrays and avereps functions. To get expression values with comparable distributions to the Affymetrix arrays, the expression values ('A' values) rather than expression ratios were used with an offset of −2.

Array probe expression values were collapsed down to single values per HGNC gene symbol by taking the mean probe expression value per gene. Gene symbols were matched to probes using the AnnotationDbi R package (version 1.46.1) using the appropriate annotation packages from Bioconductor[74].

**Bulk-tissue RNA-seq data.** RNA-seq values quantified using HTSEQ-count for the TCGA PCPG cohort were downloaded from the NCI Genomic Data Commons (GDC) website[15,75,76]. RNA-seq counts from our previous publication were used and data were processed as previously described[50]. Raw bulk-tissue RNA-seq data from this prior study is now made available under the same accession as described below for snRNA-seq.

**Merging bulk-tissue RNA-seq and microarray data.** Microarray and RNA-seq datasets were merged into a harmonized expression matrix (Supplementary Fig. 6). First, all microarray datasets were merged into a single matrix with all genes. Expression values were then quantile normalized using the normalize.quantiles function from the preprocessCore R package (version 1.46)[77]. RNA-seq counts were then quantile normalized using the normalize.quantiles.use.target function from preprocessCore with the microarray datasets' quantile distribution as the target distribution. Batch effects were removed by fitting a linear model using the samples without missing values for each gene separately with sample genotype as a covariate using the remove-BatchEffect function from the limma R package[73]. Samples without annotated genotypes were set to zero weight in the linear model. Replicate samples were removed from the analysis after batch effect removal.

**Clustering of PCPG bulk-tissue transcriptomes**
Semi-supervised clustering of the merged expression matrix was performed using ConsensusClusterPlus R package (version 1.5)[78]. Clustering was performed using a cosine distance metric with only the genes with more than three MADs above the median coefficient of variation across all samples among the genes with no missing values and expression values mean-centered. Consensus clustering was performed using the ConsensusClusterPlus function (pItem=0.7, clusterAlg = 'hc', distance = 'pearson', innerLinkage = 'ward.D2', final-Linkage = 'ward.D2', reps = 1000, maxK = 12, corUse = 'pairwise.complete.obs', seed = 1). Initial clustering attempts identified a cluster that was associated with batch (GSE19987 and GSE2841) but

not any genotypes. The cluster had no clear gene signature and significantly higher than average normalized unscaled standard errors (NUSE), so these samples were removed from the analysis before the merging process was repeated. An initial consensus cluster number of nine was chosen based on the point at which the proportion of ambiguously clustered pairs stopped changing significantly. In addition, a cluster associated with normal adrenocortical cells and mix of genotypes (C2C) was identified. Since this was a confounding factor in the batch effect removal model, the batch effect removal process was repeated a third time with samples initially assigned to cluster C2C to zero weight to improve performance. Two clusters associated with the kinase genotypes (C2A) were later merged based on their proximity in UMAP space and common genotypes, yielding a final eight PCPG clusters. UMAP values were calculated using the umap function from the uwot R package using the same distance matrix as the clustering analysis (n_epochs = 1000, min_dist = 0.1, metric = 'cosine', nn_method = 'annoy', n_neighbors = 15)[79,80].

**Projection of pseudo-bulk snRNA-seq samples into bulk-tissue UMAP.** Single-nuclei RNA-seq counts per-sample were summed to produce pseudo-bulk expression profiles. These were also included as a separate RNA-seq batch. A process for projecting new samples into the existing UMAP projection was devised to compare the clustering performance of pseudo-bulk analysis with all cells versus NEO cells only. First, pseudo-bulk samples were quantile normalized using the same quantile target distribution as used for the bulk microarray merging process. Coefficients of the new samples' batch were calculated by taking the mean value of the expression values minus the coefficients of the corresponding genotypes of the new samples from the original linear model fit. The batch coefficients were then subtracted from the quantile-normalized expression values. Any missing genes from the new samples from the highly variable gene list from the bulk clustering analysis were then imputed using k-nearest-neighbors imputation from the final bulk expression matrix by taking the mean value from the 15 nearest neighbors (by cosine distance) in the bulk matrix. The resulting expression matrix was then mean-centered and projected onto the original umap model using the umap_transform function from the uwot R package using the UMAP model from the original bulk compendium.

**Bulk-tissue differential gene expression analysis.** Bulk-tissue differential gene expression analysis was performed using the limma and edgeR packages in R. For DE analysis of bulk tissue, the quantile-normalised bulk gene expression compendium was used. Linear models were fit for genotype and tumor subtype. For the metastatic versus non-metastatic analysis, we assumed any annotation of malignant cases corresponded to a metastatic phenotype to conform to the current nomenclature. Samples that did not have clinical data for malignant/metastatic status were removed prior to modeling, and subtype and metastatic were modeled as a single model factor. Batch was included as a factor in each model to prevent confounding by batch effects.

Contrast coefficients and standard errors for each gene were estimated using the contrasts.fit function and log-fold-change values, t-statistics and corresponding $p$ values were computed by an empirical Bayes method with the eBayes function using the limma-trend method. Within each contrast, this process was repeated for blocks of groups and genes such that each gene's coefficients were calculated with all groups without missing values for that gene, as not every gene was represented in the final expression matrix by samples from every genotype or cluster group. P-values were then adjusted for multiple testing to control the false discovery rate using the BH method.

Gene set enrichment scores for bulk data were calculated using GSVA[81]. Gene sets that were used comprised the Molecular Signatures Database Hallmark gene sets[82], fetal adrenal cell-type specific gene

sets[49] (Supplementary Data 11) and gene sets derived from stromal and immune cell types using our snRNA-seq data (log2FC > 3, $P < 0.05$, Supplementary Data 13). To determine DE at the gene-set level, GSVA scores were modeled using the standard limma pipeline, with design matrices constructed as described for the metastatic vs non-metastatic analysis above.

**Pseudo-bulk differential gene-expression analysis.** Pseudo-bulk expression profiles were created at the level of broad stromal and immune cell types for each tumor and normal adrenal samples. For comparisons between tumor subtypes and normal adrenal, nuclei were removed if there were <10 from a given sample in a cluster. A small number of tumor sample nuclei that were classified as normal chromaffin cells were also removed. DE analysis was performed using the standard limma workflow. Tumor subtype and cell type were modeled as a single design matrix factor and sex was included as a second factor to prevent confounding by patient sex. Tumor and normal samples were contrasted for each subtype and cell type. Cell type-specific gene signatures were identified by performing pseudo-bulk DE, comparing each major cell type cluster (aggregated per sample) to all other non-tumor cell types (log2FC > 3, $P < 0.05$) (Supplementary Data 13). To account for ambient RNA effect, correlation between cell types from the sample of origin was estimated and sample of origin was modeled as a random effect in the data. Inspection of gene-expression between cell types within individual samples was also done to exclude any genes associated with potential ambient RNA that originated from unrelated cell types in the same sample. For instance, adrenocortical signature genes overexpressed in many normal adrenal cell types compared to the same cell types in tumor nuclei. Post-hoc filtering of adrenocortical-related genes was performed for tumor versus normal analyses.

Additional tumor versus normal tissue comparisons were performed for pseudo-bulk profiles aggregated per sample at the level of cell subtypes, where there was sufficient representation of the cell subtype (>300 nuclei) across both tumor and normal tissues (stalk-like endothelial cells, tip-like endothelial cells, macrophages) (Supplementary Data 7). Pseudobulk expression profiles for each subtype were aggregated per-sample and samples with below 50 cells were removed. Macrophages from each tumor subtype were also compared in a 1-vs-rest comparison.

Receptors and ligands specific to SCLCs were identified by comparing SCLCs to all other cells including NEO cells (log2FC > 3 and BH adjusted $p < 0.05$). Additional comparisons were performed between each non-tumor cell type (aggregated per sample) vs all other cell types, enabling identification of cell type-specific gene signatures (log2FC > 3, $P < 0.05$) (Supplementary Data 13). Heatmaps and dot plots were generated using the ComplexHeatmap R package (version 2.6.2).

**Differential cell type abundance.** The Speckle R package was used to test for statistically significant differences in cell type abundance between different tumor subtypes and genotypes. The get-TransFormedProps function was used to calculate logit-transformed cell type proportions for each tumor sample. To test for differences between the tumor genotypes, a design matrix was constructed with tumor genotype as a model factor. For the subtype comparisons, a design matrix was made with tumor subtype as a model factor. Each subtype and genotype were compared to all other subtypes and genotypes using the propeller.ttest function.

**NATMI analysis of cell–cell signaling.** Seurat normalized expression values were converted to CPMs then grouped by cell subtype and sample prior to NATMI analysis (https://github.com/asrhou/NATMI). The python 3 version of NATMI ExtractEdges with the suggested dependency versions was run on each sample with default settings.

The predicted ligand-receptor interactions were then read into R for further analysis. For a cell–cell connection to be kept for further analysis the ligand and receptor were both required to be expressed in >10 cells. Connections were also filtered out if receptor or ligand detection rate <0.1. Cluster autocrine-signaling and interactions where the ligand and receptor were the same gene were removed for data presentation and interpretation. Furthermore, interactions that were not seen in at least two samples were removed.

**Visualizing gene-expression in the TCGA pan-cancer data.** Level 3 gene expression counts were downloaded from the genomic data commons using the TCGAbiolinks R package. Raw counts were TMM normalized and transformed into log2 CPM using the edgeR R package. For the tumor comparison the log2 CPMs of *GPR139* were then plotted for samples defined as "Primary solid Tumor", "Metastatic", "Additional - New Primary" or "Recurrent Solid Tumor" by their TCGA barcode[15]. For the normal tissue comparison, log2 CPMs were plotted for samples defined as "Solid Tissue Normal" by their TCGA barcode against the previously mentioned tumor types for tumors in the PCPG cohort.

**Cell classification of PCPG NEO cells using a fetal adrenal reference.** Fetal adrenal medulla snRNA-seq data (nSamples = 17) previously generated and pre-processed by[49] was downloaded as a Seurat object from (https://adrenal.kitz-heidelberg.de/developmental_programs_NB_viz/). These data were visualized using the UMAP coordinates provided. This dataset was used as a reference to classify NEO cells and SCLCs from PCPG (nSamples = 30), normal chromaffin cells and SCLCs from adult NAM (this study). Classification was performed using the same method as described for supervised classification of stromal and immune cells using the 3000 most variable genes in the fetal adrenal data identified using Seurat FindVariableFeatures function.

**Fetal cell gene-module scoring in snRNA-seq and bulk-tissue gene-expression data.** Cell-type specific gene sets for fetal adrenal medulla cell populations were downloaded from Jansky et al. supplemental data[49] (Supplementary Data 10). Gene-signature scores for PCPG nuclei were calculated with the AddModuleScore function in Seurat with default parameters. Briefly, this function scores single cells according to the average expression of a gene-expression program and subtracts aggregated expression of a set of (100) control genes. GSVA (v1.38.2) was used to calculate gene set scores for bulk-tissue gene expression profiles of the bulk gene-expression compendium.

**RNAscope in situ hybridization.** For RNAscope® ISH, a 20ZZ probe (Hs-CDH19) targeting 456–1527 nucleotides of *CDH19* (GenBank accession NM_021153.3) was designed and manufactured by Advanced Cell Diagnostics (ACD, Newark, CA). Four-micron thick sections of formalin-fixed paraffin-embedded tumor tissue was mounted on positively charged Superfrost® Slides. The RNAscope® ISH assay was performed using an RNAscope® 2.5 HD Assay-BROWN detection kit (ACD) following the manufacturers method. Sections were deparaffinized briefly and then subjected to target retrieval using Pre-treatment 1, 2, and 3 solutions for 10 min at RT, 15 min at 100–104 °C and 30 min at 40 °C respectively, rinsing with dH$_2$O between each step. For the probe hybridization, slides were incubated with the *CDH19* probe (or the RNAscope® ISH positive control probe PPIB (Cyclophilin B) for 2 h at 40 °C in a HybEZ™ oven, followed by a series of signal amplification steps involving incubating with specific amplification solutions. For signal detection a premix DAB solution was used at RT for 10 min. Slides were counterstained with hematoxylin for 2 min, dehydrated mounted and cover slipped. Images were acquired using an Olympus BX51 fluorescent microscope (Olympus, Tokyo, Japan).

**Immunohistochemistry and scoring of immune cells.** Immunohistochemistry (IHC) using 3,3′-Diaminobenzidine (DAB) was performed on FFPE tissue sections using commercially available antibodies. Details of antibodies and antigen retrieval methods is described in Supplementary Table 4. Staining for S100 was done using a Leica-Bond-3 automated staining platform (Leica Micro systems, Mount Waverley, Victoria, Australia). All other stains were performed manually. Immune cell scoring was done by an expert pathologist (AJG). To determine inflammatory cell counts, areas of blood extravasation were avoided as much as possible as were areas of fibrosis. Only inflammatory cells within the tumor were counted (that is circulating or marginating inflammatory cells were not counted). CD206 also stains for marginating neutrophils in capillaries, which were therefore ignored. Similarly, CD206 positive neutrophils in the stroma were disregarded.

### Reporting summary

Further information on research design is available in the Nature Research Reporting Summary linked to this article.

## Data availability

The snRNA-seq as well as bulk-tissue RNA-seq data generated in this study have been deposited in the European Genome-Phenome Archive (EGA) under accession code EGAS00001005861/ EGAD00001008403. The data are available under restricted access as it is potentially identifiable based on patient genotype. Access can be obtained by researchers upon application through EGA to the study data access committee (DAC) of the University of Melbourne. The DAC will attempt to provide a response to all applications within ten days of submission and render a final decision within no more than four weeks. Once the DAC has in principle approved an application a data transfer agreement (DTA) will be mutually agreed and executed between institutions and data will then be made available through EGA. The remaining data are available within the article and supplementary information. Source data required for the reproduction of figures presented in this study are available from figshare (https://doi.org/10. 6084/m9.figshare.21080476). The publicly available microarray datasets used in this study are available from the Gene Expression Omnibus (accession numbers GSE131907[30] GSE146771[31], GSE2841[9], GSE19422[83], GSE19987[84], GSE39716[85], GSE50442[85], GSE51081[86], and GSE67066)[87] and ArrayExpress (accession number E-MTAB-733)[58]. In addition, publicly available Affymetrix Cytoscan HD array data is available from the Gene Expression Omnibus under accession numbers GSE61594[50]. and GSE94378[26]. Source data are provided with this paper.

## Code availability

The code used for data analysis is available at https://github.com/ UMCCR-RADIO-Lab/snRNA-seq-atlas-of-pheochromocytoma-and-paraganglioma.

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

## Acknowledgements

We acknowledge the Victorian Cancer Biobank for provision of patient samples. Staff from the flow cytometry facility and Centre for Advanced Histology and Microscopy at the Peter MacCallum Cancer Centre and the Clinical Genomics Platform at University of Melbourne Centre for Cancer Research for their support in generating primary data. Brian Fritz and 10x Genomics for contribution of reagents to the study. We thank Alistair Forrest and Rui Hou at the Harry Perkins Institute of Cancer Research for assistance with NATMI analysis and Alicia Oschlack for her helpful advice on data analysis. This work was supported by funds from the PMF Foundation and a National Health and Medical Research Project Grant (APP1108032, RCB, RWT). RWT was supported by a Victorian Cancer Agency Mid-Career Fellowship (TP828750). A.Pa and A.F. were supported by the Joseph Herman Trust at the University of Melbourne. K.P. was supported by the Intramural Research Program of the NICHD, NIH. P.L.M.D. is a Robert Tucker Hayes Distinguished Chair in Oncology and was supported by funds the NIH/NIGMS (GM114102), NIH/NCI (CA264248), Neuroendocrine Tumor Research Foundation. L.F. was supported in part by ACS MRSG-15-063-01. A.S.T. was supported by the Neuroendocrine Tumor Research Foundation and Paradifference Foundation.

## Author contributions

R.W.T. conceived the study. M.Z., L.M., A.P., B.B., S.B., A.F., and F.J.R. performed experiments and data analysis. A.S.T. and A.J.G. performed pathology review and analysis. A.H., J.A.M., Z.F., S.G., L.F., P.L.M.D., R.J.H., R.C.B., and K.P. provided material support, recruited patients or curated clinical data. M.Z., A.P., B.B., S.B., and R.W.T. wrote the manuscript. All authors edited and approved the final manuscript.

## Competing interests

The authors declare no competing interests.

## Additional information

Magnus Zethoven[1,14], Luciano Martelotto[2,14], Andrew Pattison[2,14], Blake Bowen[2,14], Shiva Balachander[2], Aidan Flynn[2], Fernando J. Rossello[2], Annette Hogg[1], Julie A. Miller[3,4], Zdenek Frysak[5], Sean Grimmond [2], Lauren Fishbein[6], Arthur S. Tischler[7], Anthony J. Gill [8,9,10], Rodney J. Hicks[1], Patricia L. M. Dahia [11], Roderick Clifton-Bligh [8,9], Karel Pacak [12] & Richard W. Tothill [2,13] ✉

[1]Peter MacCallum Cancer Centre, Melbourne, VIC, Australia. [2]Centre for Cancer Research and Department of Clinical Pathology, University of Melbourne, Melbourne, VIC, Australia. [3]Department of Surgery, Royal Melbourne Hospital, Parkville, VIC, Australia. [4]Department of Surgery, Epworth Hospital, Richmond, VIC, Australia. [5]3rd Department of Internal Medicine – Nephrology, Rheumatology and Endocrinology, Faculty of Medicine and Dentistry, Palacky University Olomouc and University Hospital Olomouc, Olomouc, Czech Republic. [6]Department of Medicine, Division of Endocrinology, Metabolism, Diabetes, University of Colorado, Aurora, CO, USA. [7]Tufts Medical Centre, Boston, MA, USA. [8]Sydney Medical School, University of Sydney, Sydney, NSW, Australia. [9]Kolling Institute of Medical Research, Royal North Shore Hospital, Sydney, NSW, Australia. [10]NSW Health Pathology, Department of Anatomical Pathology, Royal North Shore Hospital, Sydney, NSW, Australia. [11]Div. Hematology and Medical Oncology, Department of Medicine, Mays Cancer Center, University of Texas Health Science Center at San Antonio (UTHSCSA), San Antonio, TX, USA. [12]Eunice Kennedy Shriver National Institute of Child Health and Human Development, Bethesda, MD, USA. [13]Sir Peter MacCallum Department of Oncology, University of Melbourne, Melbourne, VIC, Australia. [14]These authors contributed equally: Magnus Zethoven, Luciano Martelotto, Andrew Pattison, Blake Bowen. ✉e-mail: rtothill@unimelb.edu.au

