## [Peer Review File · Nature Communications]

Title: Single-nuclei and bulk-tissue gene-expression analysis of pheochromocytoma and paraganglioma links disease subtypes with tumor microenvironmentREVIEWER COMMENTS

Reviewer #1, expert in neuroendocrine tumours and single-nuclei RNA-seq (Remarks to the Author):

Zethoven et al used single nuclei-RNA-seq to dissect cell composition, refine PCPG subtypes and compare PCPG with normal tissue expression. They have addressed this problem with snRNA-seq of human PPGL, and published non-malignant fetal adrenal gland tissue. They validate some of their observations with RNA-FISH.

The bioinformatics analysis is rigorous and largely complete. The strength of the manuscript is that it shows how single-cell analysis can shed light on tumor microenvironment, especially how the tumor microenvironment is dominated by a pro-angiogenic cell types.

The largest caveat of the study is that the authors have a rather strong batch effect on neoplastic cells, that cluster according to patient instead recapitulating a clustering pattern as previously described from bulk analysis (hypoxia group, kinase group).

I would ask the authors to consider the following:

(1)

In Fig1 and text (line 128)

The author state and show in Fig 1:

“Stromal and immune cell nuclei clustered by cell lineage, indicating minimal technical variability, whereas neoplastic (NEO) nuclei clustered by sample of origin.

As indicated already in the introduction that PPGL come in different groups, clustering of neoplastic cells by UMAP was not congruent with this clustering patterns as previously described using bulk-tissue gene-expression profiling.

This indicates that there a rather high technical variability within the neoplastic cells?

Can the author overcome this by controlling better for batch effect?

The alternative approach using pooling transcripts from NEO cells to simulate bulk tissue RNA-seq data (pseudo-bulk) followed by multi-dimensional scaling (MDS) is not a convincing approach to overcome this problem.

(2)

Regarding txt 306-308 :

“Whether sustentacular cells in PCPG are precursors or terminally differentiated cannot be confirmed in our data, therefore we described them as Schwann-cell-like rather than SCPs.”

Since SCP are embryonic precursors of chromaffin cells, the authors should clarify if sustentacular glia cells are neoplastic or stroma.

InferCNV for instant could help to get some understanding if glia cells are neoplastic or stroma. InferCNV is often used to explore tumor single cell RNA-Seq data to identify evidence for large-scale chromosomal copy number variations. The authors already indicated in Figure 2F inferred copy-number in NEO and

non-NEO cell types by gene-expression for an SDHB-associated PGL. Are schwann cells belonging to stroma or Neo in that sample?

Also, using CNV profiles from array-CGH data from the same tumors allows to derive a combined score that quantifies the extent to which the expression pattern of a given cell reflects the CNV profile of a given patient.

IHC combined with Fish might also further validate if glia belongs to stroma or neoplastic cells.

(3)

Comparative analysis with healthy tissue is important to make conclusion to what extend PCPG exhibit variable chromaffin cell differentiation patterns during development.

The authors state: (txt 336-337)

“PCPG variably express chromaffin-related genes indicating divergent states of cellular differentiation or developmental origins. To determine the similarity of PCPG to sympathoadrenal cells during early development we used a published snRNA-seq dataset of normal human fetal adrenal tissues at seven developmental time points”

Jansky et al used fetal developing adrenal gland to understand if childhood neuroblastoma is linked to embryonic development.

However, SCP are rather short-lived during embryogenesis, and PPGL are never observed in children, mostly adults.

It would be interesting to include comparative analysis with a postnatal adrenal gland tissue, that also has been recently published and provided insight to postnatal developmental programs in the adrenal medulla.

Reviewer #2, expert in neuroendocrine tumours (Remarks to the Author):

The authors have done their best to explore three main issues in this manuscript; the single cell mRNA sequencing of 32 frozen PCPG tissues, bulk gene expression analysis of published microarray and mRNA seq data of more than 700 tissues; and looked at immune cell subtypes in PCPG tissues.

My major concern is that of the 32 tissues used for single cell mRNA seq, no tumor cell percentage is provided. Further on in the manuscript, the authors mention a broad range in the relative contribution of stromal and immune cells. this should not be surprising, given the fact that whole fresh frozen tumor sections (biopsies) were the starting point. It thus becomes very hard to draw conclusions on the contribution of specific stromal and immune cell subtypes, as the certain populations can be over- or underrepresented based on where the resection margin was.

The clear benefit of single cell analysis is the ability to look at tumor cell heterogeneity. The authors

however provide only minimal data on this, and quickly go on to pseudo-bulk analysis and comparison to published bulk expression data. The authors then go on to further differentiate the already known PCPG clusters into subtypes based on specific driver gene expression.

the significance to the field lies mainly in the further differentiation of PCPG subtypes, but as correlation with clinical outcome is lacking, and no external validation has been performed, one has to wonder what the clinical significance of these subtypes is. The significance of the single cell mRNA seq analysis is very limited; subclonal cell populations are seen but not further explored.

Altogether, I think this manuscript would benefit from either further exploration of the single seq data; or further validation of the new PCPG subtypes (by correlation with clinical parameters and/or external validation).

As discussed with the editor I have not formally reviewed the part of the manuscript on immune and stromal cell subsets.

Reviewer #3, expert in immune cell types/TME/neuroendocrine tumours (Remarks to the Author):

The authors have used the innovative and emerging technology based on single nuclei-RNA-seq (snRNA-seq) for analysis of a cohort of pheochromocytomas and paragangliomas (PPGL) in order to identify the transcriptional profile of PPGL regarding multiple pathways related with PPGL and their respective clusters, as well as to further investigate the composition of the PPGL microenvironment, particularly in terms of the PPGL-associated immune cells and stromal cells.

The paper contains a massive amount of transcriptomic data, suggesting some novel pathways as well as several already known pathways associated with PPGL, and is well-written. The study itself is well-conducted from a methodological point of view, and also very relevant for the field, but there are two main shortcomings:

- 1) lack of validation of the most important findings using other methodologies; for instance, the composition of the immune microenvironment could be further validated with flow cytometry or immunohistochemistry experiments, or even using RNA scope validating some recognized markers expressed by the immune or stromal cells.
- 2) low number of cases included: considering the wide heterogeneity of the genetics related with PPGL and the number of cluster subgroups, as well as the heterogeneity of the tumor microenvironment and related pathways such as angiogenesis, the size of the studied cohort for this kind of analysis is relatively small, with some subgroup clusters containing very few cases for meaningful conclusions. Moreover, there were only 3 metastatic cases, and only 2 locally invasive PPGL, which does not allow to investigate properly the microenvironment and other pathways for the most aggressive cases, which are the ones likely having increased expression of some of the pathways explored and likely more “immunologically” hot. Also, including these few aggressive cases with non-aggressive PPGL may skew or deviate the transcriptional profile and the microenvironment composition and somewhat mislead the whole

analysis by cluster.

Additionally, it was surprising to me that normal adrenal medulla appeared to contain more prominently macrophages belonging to the M2-like subtype. Indeed, it would be perhaps expected that more M2 macrophages were present in the PPGL in comparison to normal adrenal medulla. Usually M2-macrophages are considered pro-tumorigenic, and in some other neuroendocrine tumors (like in pituitary) it seemed that M2-macrophages predominate, while M1-macrophages were more predominant in normal tissues (many normal tissues don't even have any M2-macrophages). Could the authors explain this? Perhaps this is because of the relatively simple assumption based on only two markers analysed for M2-macrophages – could the authors run the analysis focused on other relevant macrophage markers such as CD206, Oxid Nitric, Arginase-1, CD86, HLA-DR1, etc... in fact, the M2-M1 polarization theory is limited and the macrophages should be more accurately regarded as part of a phenotypic continuum, hence a more thorough phenotypic characterization is often required. Hence, postulations of M1 or M2-like macrophages are more accurate with better phenotypic characterization, including looking for markers of M1 macrophages (HAD-DR, Oxid Nitric, etc). Eventually, wouldn't be better naming in the paper CD163+-macrophages and MARCO+-macrophages, rather than M2 macrophages?

Reviewer #4, expert in multi-omics/bioinformatics (Remarks to the Author):

Summary:

In this manuscript, Zethoven et al. conducted an integrative analysis of snRNA-seq and bulk RNA-seq datasets of patients with pheochromocytomas and paragangliomas (PCPG) with diverse disease conditions and genetic backgrounds. Being one of the first studies focused on single-cell dissection of the PCPG microenvironment, this study has the potential to advance our understanding of the molecular and cellular dynamics of this rare and under-studied neuroendocrine cancer type. However, the reviewer finds the valuable snRNA-seq resources generated in this study underutilized, yielding largely confirmatory results rather than novel insights that are unseen in the previous bulk studies. In addition, the lack of rigorous statistical testing and the lack of depth in the analysis throughout the manuscript, which often leads to indefinite and confusing conclusions, undermines the overall rigor of this study and clearly calls for a major overhaul before being considered for publication in Nature Communications.

The last section contains arguably some of the most solid and innovative analyses in this study in that the authors were able to bridge tumor progression with organogenesis, particularly in the fetal stage, which is an important and fascinating topic given the recently booming literature on the oncofetal characteristics of human cancers.

When it comes to data transparency, the authors have done a laudable job of providing all code on a publicly accessible repository and making the raw data available on EGA. These efforts demonstrated the authors' commitment to open science and the advancement of community knowledge.

Major comments:

1. Regarding the analyses in the second section on defining transcriptional subtypes of PCPG, it is unclear what the contributions of the snRNA-seq data are. Most results shown here are obtained from the analysis of public bulk RNA-seq datasets. The snRNA-seq data seems only to be playing a role of validation, which is a clear let down. Specifically, removing all the squares (denoting snRNA-seq pseudobulks) from figure 2b will have no effect on the fundamental clustering pattern formed solely by the dots (denoting real bulks), meaning that the separation between the PCPG subtypes will still stand.

The reviewer does not intend to dismiss the efforts made by the authors on collecting and harmonizing public large-scale PCPG RNA-seq datasets, which has yielded meaningful results of PCPG molecular subtyping through an integrative analysis. Instead, the reviewer encourages the authors to further devise innovative analytical schemes that make the most use of the snRNA-seq data on top of the bulk atlas.

2. The conclusion of the PCPG tumor microenvironment being “pro-angiogenic”, as presented in the third section, is weak due to multiple flaws in the analyses.

First, the authors did not conduct any statistical analysis (or at least did not show any significance numbers or indicators) in comparing cell type fraction or gene expression between conditions or subtypes (the stats hidden in the supplementary tables do not count). For instance, the claim in line 252 that “..., consistent with there being a higher number of vascular cell types in VHL-mutant PCPG” could use a t-test on the abundance of the vascular cell types between VHL PCPG samples versus the others.

Second, the clustering patterns in figures 3c and 3d are suspicious. Across a wide range of single-cell cancer studies, one common observation is that only malignant cells show clear separation between tumor and normal tissues. Even if there are authentic biological differences in fibroblast or endothelial transcriptome landscape between tumor and normal tissues, the authors failed to reveal any such signal. For example, the authors did a DE analysis between all the tumor fibroblasts against all the normal fibroblasts and found that the top DE genes are highly expressed in a cell type (myofibroblast) that only exists in the tumor compartment (lines 233-235). This circular argument does not help with clarifying the underlying gene expression structures that distinguish specific fibroblast subtypes in the tumor context from their counterparts in the normal context. Thus, what the authors should have done is to only include a single fibroblast or endothelial subtype each time for the DE analysis.

Third, still regarding the fibroblast and endothelial sub-clustering results, there is heterogeneity even within the same cell subtype of the same tissue compartment. Specifically, one can see that the “Stalk-like ECs” form two clusters in the tumor endothelial space. This type of unusual clustering pattern needs further investigation from the authors. If this variation turns out to be due to batch effect or other non-biological confounding factors, the authors need to revise the entire clustering strategy and take extra care when interpreting these clustering results.

3. The analyses on the immune compartment of PCPG are lackluster and unreliable. Importantly, the major premises of the entire section that “Immune cell infiltrates in PCPG are predominantly macrophages” (line 267) and that “a low lymphocytic infiltrate is consistent with PCPG being immunologically cold” (line 299) are in direct conflict with a previous scRNA-seq study on PCs (PMID: 34905486) in which the most abundant cell types are in fact of the T cell lineage, followed by endothelial cells, myeloid cells, fibroblasts, and others. A widely appreciated drawback of snRNA-seq in comparison to scRNA-seq is that the former often sub-samples T cells while retaining comparable myeloid abundance. Thus, without independent validation from an orthogonal data mode derived from the same set of samples (e.g. IHC, CyTOF, etc.) the authors simply cannot claim either that the PCPG microenvironment is dominated by macrophages or short of immune infiltration.

Additionally, the authors stayed on the very surface when analyzing the immune compartment of PCPG in terms of failing to reveal specific and often times well-known immune cell states and their interactions with PCPG subtypes or other disease features. The most conspicuous case supporting this argument is that the authors did not show any cell state diversity within the macrophage population and only mentioned a few M1 or M2 gene markers that seem to be differentially expressed between tumor and normal. This problem also applies to the analysis of the T cell population where the authors only managed to annotate cytotoxic CD8+ T cells out of the entire CD8+ T cell population and Treg out of the entire CD4+ T cell population, which is clearly not up to the current standard of single-cell immune analysis.

Lastly, the authors failed to interpret appropriately what the differential expressions of certain immune cell markers actually mean in the context of bulk samples (similarly in figure 3g). For example, it seems that all four macrophage markers (CD163, MARCO, PLAU, and CXCL2) are more abundant in C1A2 compared to the other subtypes. But does it mean that C1A2 is of a more immuno-suppressive myeloid scenario or just that there are overall more macrophages in C1A2?

4. The issues described above, such as lack of formal statistical testing, arbitrary selection of gene markers as proxy of cell states/populations, failure in disentangling cell type abundance and cell type specific gene expression in analyzing bulk samples, all apply to the fifth section (“Schwann-cell-like cells (SCLCs) and putative paracrine signaling with NEO cells”) as well.

5. Although the authors were able to show that GPR139 expression is restricted to C1 PCPG, it may not bear a strong potential for PCPG therapeutics as anticipated by the authors. This is mainly due to the fact that GPR139 RNA expression is observable in fetal and adult neurons (see GTEx, Descartes, and Human Protein Atlas) and that GPR139 shows high antibody staining in normal adrenal gland granular cells, which makes it a non-exclusive marker of PCPG tumor cells.

Minor comments:

1. Eight PCPG gene-expression subtypes were determined in the analysis besides the normal subtype, but the authors claimed only seven in the abstract.

2. The term SCLC first appeared in figure 1d while its definition was given later in line 225.
3. No statistical tests were performed for analyses matching figures 5a-b, thus the claims regarding differential cell type abundance between subtypes are invalid.
4. There is a typo in figure 1h: the y axis should be “dim 2” instead of “dim 1”.
5. There is a typo in Table S2: Fibroblast > Fibroblasts. Also, it seems what is named as “Smooth muscle cells” has taken the other name of “ACTA2pos fibroblasts” in this table. The authors need to adopt a unified naming scheme.
6. Why did the authors say “Two consensus clusters were consolidated...” (line 173) while there are nine in figure 2c?
7. There is a typo in the title of the third section: remove the indefinite article “a”.
8. Human reference genome hg19 is outdated. Mapping reads onto this genome is not the best practice although it is not expected to produce highly skewed results compared to mapping onto hg38.

26th July 2022

We thank the reviewers for their valuable feedback on our manuscript.

We have detailed our responses to the individual reviewers below including additional analysis and experimentation as required with the respective changes also made to the manuscript, Figures and Supplementary data.

Reviewer #1

Comment: The largest caveat of the study is that the authors have a rather strong batch effect on neoplastic cells, that cluster according to patient instead recapitulating a clustering pattern as previously described from bulk analysis (hypoxia group, kinase group).

Can the author overcome this by controlling better for batch effect?

Author response: Our expectation of a batch effect would relate to a non-biological technical artefact introduced during experimentation. We believe the best internal control is the normal stromal and immune cells and given the normal cell types including normal chromaffin cells from two independent adrenal tissues co-clustered based on the cell lineage regardless of the sample of origin or experimental batch we believe this would argue against a major batch effect in our data (see **Supplementary Figure S2**).

Furthermore, despite our general observation of NEO cells clustering by their sample of origin, this is not always the case. For instance, we observed tight co-clustering of NEO cells from two synchronous primary PG tumors (PGL1 and PGL3) (see **Supplementary Figure S5e**). At the genotype level NEO cells from *SDHD*, *MAX*, *NF1* and *MAML3*-fusion PCPG also clustered in close proximity based on their respective genotype (**Supplementary Figure S5a-d**) and while a perfect separation of NEO cells based on PCPG C1 and C2 subtypes was not observed, there was clear evidence of clustering of tumors by PCPG subtype in the UMAP (**Figure 1h**).

Despite the evidence to suggest no significant batch effect in the data we specifically addressed the reviewer's request to use batch correction using methods such as *Seurat* standard integration (Canonical Correlation Analysis), *Seurat* Reciprocal PCA integration, and *Harmony* integration. Only the *Harmony* method retained the expected normal stromal and immune cell clustering pattern by UMAP; however, by removing patient specific effects the expected biological structure in the data in relation to PCPG subtype and genotype was also removed. In the *Harmony* corrected data nearly all NEO cells formed a single NEO cluster in the UMAP space with no separation of C1 and C2 tumors. The lack of distinct NEO cell clusters in the *Harmony* normalised data indicates that the data was being overcorrected. It was not clear to us what purpose this overcorrected data could serve for interpreting the biology of the cancers. We have added reference to the *Harmony* batch correction in the manuscript and have included results in **Supplementary Figure S3**.

The alternative approach using pooling transcripts from NEO cells to simulate bulk tissue RNA-seq data (pseudo-bulk) followed by multi-dimensional scaling (MDS) is not a convincing approach to overcome this problem.

Author response: We have now removed the MDS plot and reference to this analysis from the manuscript as we felt it was unnecessary for defining subtypes and may have been confusing to the reader.

(2) Regarding txt 306-308 :Since SCP are embryonic precursors of chromaffin cells, the authors should clarify if sustentacular glia cells are neoplastic or stroma. InferCNV for instant could help to get some understanding if glia cells are neoplastic or stroma. InferCNV is often used to explore tumor single cell RNA-Seq data to identify evidence for large-scale chromosomal copy number variations. The authors already indicated in Figure 2F inferred copy-number in NEO and non-NEO cell types by gene-expression for an SDHB-associated PGL. Are schwann cells belonging to stroma or Neo in that sample?

Author response: We applied InferCNV to all samples (**Supplementary Figure S4**) and have now included additional InferCNV plots for two PCPG tumors that had large number of SCLCs (**Figure 5b**). Almost all SCLCs clustered together and had a diploid profile. We cannot exclude the possibility that a very small number of SCLCs in E042 had an aneuploid profile based on the InferCNV analysis but clustering of a small number of SCLCs with NEO cells is likely an anomaly due to imperfect clustering as we noted that a small number of tumor cells also clustered among the diploid SCLC cells as shown for P018-PGL3 as shown in **Figure 5b**.

Also, using CNV profiles from array-CGH data from the same tumors allows to derive a combined score that quantifies the extent to which the expression pattern of a given cell reflects the CNV profile of a given patient.

Author response: We had matched Affymetrix SNP array data for some cases as shown in **Figure 1f** and **Figure 5b**. In the tumors without matched SNP array data the recurrent chromosome arm-level deletion events expected in this disease were frequently observed in the InferCNV profile of the tumors.

We are not familiar with the specific method the reviewer refers to that would quantify the extent to which the transcriptional profile reflects the CNV profile of the tumor. However, expect that the underlying CNV profile of a NEO cells may in part explain why NEO cells from different tumor samples often cluster independently by UMAP as well as patient specific effects.

IHC combined with Fish might also further validate if glia belongs to stroma or neoplastic cells.

Author response: As stated in the manuscript we cannot exclude the possibility that a minor subpopulation of aneuploid SCLCs existed in PCPG tumors but in our opinion the evidence for this is quite limited. Most if not all SCLCs had a diploid profile. This is entirely consistent with previous studies using orthogonal IHC and flow cytometry methods as we cited in the manuscript (p11, line 377). Although it would be remarkable to find a subpopulation of aneuploid SCLCs/SCPs in PCPG we do not think sufficient evidence exists to warrant further validity studies by FISH/IHC and this would be out of scope of the current study.

It would be interesting to include comparative analysis with a postnatal adrenal gland tissue, that also has been recently published and provided insight to postnatal developmental programs in the adrenal medulla.

Author response: We included two (post-natal) adult normal adrenal medulla (NAM) tissues in our 10x sn-RNA-seq dataset (E240 and E243). In the NAM data we identified SCLCs with a transcriptional profile similar to the SCPs described by Jansky at colleagues. These SCLC cells exhibited high expression of canonical SCPs markers including SOX10 (**Figure 1e**), and high GSVA scores for early, late, and cycling SCPs as defined by Jansky et al (see **Figure 6b, c,d**). We note that sustentacular cells that express SOX10 and S100 are also a well-known features within normal adult adrenal medulla tissues and PCPG tumors. We believe SCLCs identified in our analysis are highly likely to be sustentacular cells.

As requested we have **also** compared our post-natal NAM 10x sn-RNA-seq data to the post-natal adrenal data described by Bedoya-Reina and colleagues (<https://doi.org/10.1038/s41467-021-24870-7>) Bedoya-Reina used Smartseq2 scRNA-seq to identify two adrenal medulla cell types in the postnatal adrenal medulla tissue: 1) chromaffin cells and 2) cholinergic progenitor cells. It is worth noting that Bedoya-Reina et al did not identify SCLCs/SCPs (glial cells) in postnatal adrenals, in contrast to our observations. The absence of SCPs/SCLCs might be explained by Smartseq2 platform having lower cell-throughput compared to 10x platform. We found in our data that SCLCs were only 1.3-2.2% of all cells in post-natal NAM tissues (**Supplementary Table S2**) and therefore very few SCPs/SCLCs were likely captured by Smartseq2 explaining why these cells were not identified by UMAP or tSNE clustering described by Bedoya-Reina.

Bedoya-Reina et al identified gene sets for chromaffin cells and cholinergic progenitor cells. To determine whether these cell populations were present in our post-natal NAM data we calculated gene module scores for individual chromaffin and cholinergic progenitor cell types in our data. Normal chromaffin cells and PCPG NEO cells in our data had high gene module scores for the Bedoya-Reina chromaffin cell gene set (see **Figure R1 below**). In contrast all normal chromaffin cells, SCLCs, and all PCPG NEO cells had very low scores for the cholinergic progenitor gene signature score.

Figure R1. Module scores for chromaffin and cholinergic progenitor signature gene sets as described by Bedoya-Reina et al in NAM chromaffin cells, SCLCs and PCPG NEO cells from different subtypes.

We therefore cannot validate the presence of the cholinergic progenitors within our own post-natal NAM dataset. Furthermore, no PCPG NEO cells resembled the cholinergic progenitor cells. Given the technical challenges in comparing across single cell RNA-seq platforms and potential differences in representation of cell types within the respective datasets we do not see a strong rationale to include the Bedoya-Reina comparison in our manuscript.

Reviewer #2, expert in neuroendocrine tumors (Remarks to the Author):

My major concern is that of the 32 tissues used for single cell mRNA seq, no tumor cell percentage is provided.

Author response: The percentage of NEO (tumor) and normal cells is detailed in **Supplementary Table S2**. We have also now also added panel **Figure 1f** to visualise the tumor and major normal cell fractions in each tumor sample.

Further on in the manuscript, the authors mention a broad range in the relative contribution of stromal and immune cells. this should not be surprising, given the fact that whole fresh frozen tumor sections (biopsies) were the starting point. It thus becomes very hard to draw conclusions on the contribution of specific stromal and immune cell subtypes, as the certain populations can be over- or underrepresented based on where the resection margin was.

Author response: We agree with the reviewer that the proportion of NEO, stromal and immune cells would be dependent on tumor sampling. The tumor percentage is high for most cases as we have shown with exception of the VHL tumors, where we expected a large contribution of stromal and vascular cell types. We believe a more extensive spatial analysis would be outside the scope of the current study.

The clear benefit of single cell analysis is the ability to look at tumor cell heterogeneity. The authors however provide only minimal data on this, and quickly go on to pseudo-bulk analysis and comparison to published bulk expression data.

Author response: InferCNV analysis of PCPG tumors identified subclonal populations within some cases and we have provided examples of those cases showing clear evidence of intratumoral heterogeneity (**Figure 1g, Supplementary Figure S4**). We agree that in future studies it would be worth exploring intratumoral heterogeneity in the context of metastatic PCPG, where one may look for a subclones in a primary tumor giving rise to a metastatic clone. However, this would require matched primary and metastatic tumor samples and these samples were not available for the current study.

Altogether, I think this manuscript would benefit from either further exploration of the single

seq data; or further validation of the new PCPG subtypes (by correlation with clinical parameters and/or external validation).

Author response: We agree that understanding clinical associations with subtypes is an important question. We therefore performed additional statistical analysis to investigate PCPG subtypes and any association with the metastatic phenotype. We used available clinical data for the bulk gene-expression dataset representing 330 tumors annotated as non-metastatic and 52 tumors annotated as metastatic. A statistical association between metastatic phenotype and PCPG subtype was determined using Fisher’s exact test. As shown in **Table R1** below, subtypes including C2B2 (MAML3), C1A1 (SDHx) and C1A2 (SDHx-HN) were all positively associated with metastases (Fisher exact Bonferroni-Hochberg adjusted $p < 0.05$) (see also **Supplementary Table S5**). These results are now described in section 2 of the results. Metastatic annotation for samples aligned with respect to PCPG subtypes is also now shown in **Figure 2d**.

Table R1. Association between malignancy and PCPG subtype

Subtype	Metastatic	Non-metastatic	Odds ratio	P-value	BH-adjusted P-value
C1A1 (SDHx)	25	36	6.75829694	3.53E-09	2.83E-08
C1A2 (SDHx-HN)	11	9	9.35158792	6.41E-06	2.56E-05
C1B1 (VHL)	8	75	0.65387823	0.37374387	0.37374387
C1B2 (EPAS1)	0	21	0	0.09347803	0.10683204
C2A (Kinase)	8	163	0.22405002	1.81E-05	4.82E-05
C2B1 (MAX)	2	49	0.24338219	0.04639758	0.06186344
C2B2 (MAML3)	8	9	6.44928038	5.81E-04	0.00116171
C2C	2	56	0.20894738	0.0134466	0.02151455

Reviewer #3, expert in immune cell types/TME/neuroendocrine tumors (Remarks to the Author):

1) lack of validation of the most important findings using other methodologies; for instance, the composition of the immune microenvironment could be further validated with flow cytometry or immunohistochemistry experiments, or even using RNA scope validating some recognized markers expressed by the immune or stromal cells.

Author response: We have now done IHC for 12 cases matched to snRNA-seq data plus two normal adrenal tissues to validate immune cell infiltrates in the tumors and normal tissue. These results have been added to section 4 of the manuscript and shown in **Figure 4e** and **Table S9**.

2) low number of cases included: considering the wide heterogeneity of the genetics related with PPGL and the number of cluster subgroups, as well as the heterogeneity of the tumor microenvironment and related pathways such as angiogenesis, the size of the studied cohort for this kind of analysis is relatively small, with some subgroup clusters containing very few cases for meaningful conclusions.

Author response: We agree that profiling a larger number of cases by snRNA-seq would be ideal but given the current expense of the platform and the rarity of the PCPG tumor tissues this is not currently feasible.

It is worth highlighting the value of the large compendium of bulk RNA data representing 735 PCPG tumors used in our analysis that enabled validation of observations made from the snRNA-seq data. For example, we found evidence of increased vascular cell types in the C2B₂ (MAML3) group based on relative expression EC markers (*FLT1*, *ANGPT2*, *HEY1*, *DLL1*) supporting our highly novel observation of elevated HIF pathway activity in MAML3-fusion tumors.

Moreover, there were only 3 metastatic cases, and only 2 locally invasive PPGL, which does not allow to investigate properly the microenvironment and other pathways for the most aggressive cases, which are the ones likely having increased expression of some of the pathways explored and likely more “immunologically” hot. Also, including these few aggressive cases with non-aggressive PPGL may skew or deviate the transcriptional profile and the microenvironment composition and somewhat mislead the whole analysis by cluster.

Author response: Using clinical annotation available for the bulk RNA data we have now performed DE analysis (genes and GSEA gene-sets) contrasting metastatic and non-metastatic tumors (**Figure 8, Supplementary Figure S12, Supplementary Table S14**). A description of these results has been added in the final results section of the manuscript.

It is worth noting that we did not observe a significant difference in the expression of canonical T cell gene markers (e.g., *CD3E*, *CD8A*) between metastatic and non-metastatic PCPG; however, we did observe modest overexpression of some macrophage markers (e.g., *CD68*, *MARCO*) in metastatic C1A (SDHx) PCPG. We believe that PCPG are generally immunologically cold with respect to an adaptive immune response, but clearly further histological analysis may be required to draw any final conclusions in this regard.

*Additionally, it was surprising to me that normal adrenal medulla appeared to contain more prominently macrophages belonging to the M2-like subtype. Indeed, it would be perhaps expected that more M2 macrophages were present in the PPGL in comparison to normal adrenal medulla. Usually M2-macrophages are considered pro-tumorigenic, and in some other neuroendocrine tumors (like in pituitary) it seemed that M2-macrophages predominate, while M1-macrophages were more predominant in normal tissues (many normal tissues don't even have any M2-macrophages). Could the authors explain this? Perhaps this is because of the relatively simple assumption based on only two markers analysed for M2-macrophages – could the authors run the analysis focused on other relevant macrophage markers such as *CD206*, *Oxid Nitric*, *Arginase-1*, *CD86*, *HLA-DR1*, etc... in fact, the M2-M1 polarization theory*

is limited and the macrophages should be more accurately regarded as part of a phenotypic continuum, hence a more thorough phenotypic characterization is often required. Hence, postulations of M1 or M2-like macrophages are more accurate with better phenotypic characterization, including looking for markers of M1 macrophages (HAD-DR, Oxid Nitric, etc). Eventually, wouldn't be better naming in the paper CD163+-macrophages and MARCO+-macrophages, rather than M2 macrophages?

Author response: As suggested by the reviewer, we have visualised expression of an extended set of relevant M1 and M2 macrophage marker genes (see **Figure 4c**), revealing a heterogenous expression pattern across individual tumors and normal adrenal tissues.

IHC and bulk-tissue gene-expression indicates that there are likely to be more macrophages in C1B₁ (VHL) and potentially also C1A2 (SDHx) PCPG tumors. Furthermore, most macrophages in these tumors expressed the M2 markers CD163 and CD206 by IHC. These details have been added to section 4 of the manuscript.

IHC shows that most macrophages in normal medulla also expressed CD163 and occasionally CD206 (MRC1) consistent with M2 phenotype. However, IHC staining of normal adrenal tissues also showed more CD163+ and CD206+ cells in the adrenal cortex, which is likely contributing to most of the macrophage gene expression signal in bulk normal adrenal tissue and the respective macrophages from normal adrenals in the snRNA-seq data.

Reviewer #4

Major comments:

1. Regarding the analyses in the second section on defining transcriptional subtypes of PCPG, it is unclear what the contributions of the snRNA-seq data are. Most results shown here are obtained from the analysis of public bulk RNA-seq datasets. The snRNA-seq data seems only to be playing a role of validation, which is a clear let down. Specifically, removing all the squares (denoting snRNA-seq pseudobulks) from figure 2b will have no effect on the fundamental clustering pattern formed solely by the dots (denoting real bulks), meaning that the separation between the PCPG subtypes will still stand.

The reviewer does not intend to dismiss the efforts made by the authors on collecting and harmonizing public large-scale PCPG RNA-seq datasets, which has yielded meaningful results of PCPG molecular subtyping through an integrative analysis. Instead, the reviewer encourages the authors to further devise innovative analytical schemes that make the most use of the snRNA-seq data on top of the bulk atlas.

Author response: Given the limited biological replicates in the snRNA-seq analysis, it was important to confirm the subtype associations of samples analysed by snRNA-seq. Aside from redefining the molecular subtypes of PCPG, we wanted to prove that bulk gene-expression subtypes corresponded to the NEO cells and that individual subtypes are not overly influenced by the contribution of stromal, immune or other cell types. This is a very common

problem in the discovery of cancer gene-expression subtypes, where gene-expression signatures can be dictated by non-cancerous cells that may or may not have biological or clinical meaning. Case example in PCPG is the C2C subtype that is clearly associated with normal adrenal cortex contamination and therefore we believe should be considered a false subtype.

Integration of our snRNA-seq analysed samples with bulk-tissue data was also important for classification of some rarer genotypes that had not been extensively investigated by gene-expression profiling. This included the FH-deficient PCPG tumors. Admittedly, the importance of these small observations may not seem that significant to readers outside of the field. We agree that our analysis in section 2 is not overly sophisticated; however, we believe it was effective for the intended purpose, as described above. We have used more standard approaches for analysis of snRNA-seq data in other sections of the paper and we believe section 2 stands alone as a significant body of work for PCPG subtype validation.

2. The conclusion of the PCPG tumor microenvironment being “pro-angiogenic”, as presented in the third section, is weak due to multiple flaws in the analyses.

First, the authors did not conduct any statistical analysis (or at least did not show any significance numbers or indicators) in comparing cell type fraction or gene expression between conditions or subtypes (the stats hidden in the supplementary tables do not count). For instance, the claim in line 252 that “..., consistent with there being a higher number of vascular cell types in VHL-mutant PCPG” could use a t-test on the abundance of the vascular cell types between VHL PCPG samples versus the others.

Author response: Using the *propeller* R package, we have performed statical analysis of cell type abundance. These results have now been added to Supplementary Table S8. A T-test was performed to test differential cell type abundance of cell types in *VHL*-mutant PCPG compared to all other tumors (see **Table R2**). This confirmed that vascular cell types, comprising endothelial cells and fibroblasts were significantly more abundant in C1B1 (*VHL*) PCPG compared to other PCPG subtypes. Furthermore, overexpression of EC markers in *VHL* subtype is validated by expression in the bulk gene-expression data **Figure 3g**.

Table R2. Propellor analysis testing for the abundance of cell types detected by snRNA-seq between PCPG subtypes.

Cell type	FDR	Contrast
Endothelial cells	0.001670186	C1_vs_C2
Fibroblasts	0.046856192	C1_vs_C2
Endothelial cells	9.08E-04	C1B1_VHL_vs_rest
Tip-like ECs	0.004091911	C1_vs_C2
Stalk-like ECs	0.013985285	C1_vs_C2
Smooth muscle cells	0.03937582	C1_vs_C2
Tip-like ECs	0.001330555	C1B1_VHL_vs_rest
Stalk-like ECs	0.019627622	C1B1_VHL_vs_rest
Lymphatic ECs	0.021699369	C1B1_VHL_vs_rest

Pericytes	0.02662072	C1B1_VHL_vs_rest
-----------	------------	------------------

Second, the clustering patterns in figures 3c and 3d are suspicious. Across a wide range of single-cell cancer studies, one common observation is that only metastatic cells show clear separation between tumor and normal tissues. Even if there are authentic biological differences in fibroblast or endothelial transcriptome landscape between tumor and normal tissues, the authors failed to reveal any such signal. For example, the authors did a DE analysis between all the tumor fibroblasts against all the normal fibroblasts and found that the top DE genes are highly expressed in a cell type (myofibroblast) that only exists in the tumor compartment (lines 233-235). This circular argument does not help with clarifying the underlying gene expression structures that distinguish specific fibroblast subtypes in the tumor context from their counterparts in the normal context. Thus, what the authors should have done is to only include a single fibroblast or endothelial subtype each time for the DE analysis.

Author response: We note that despite clear separation of fibroblasts and ECs by normal and tumor tissue origin, we did not see the same effect in other stromal (e.g., SCLCs) and immune cell types. This would argue against clustering driven by a technical bias in the data as we would expect to see a similar effect across all cell types when comparing tumor and normal tissues.

To specifically address the reviewer's question, we attempted DE analysis within specific EC subsets. We could only perform this analysis for ECs including tip-like or stalk-like ECs as these cell populations were abundant in both tumor and normal tissues, whereas fibroblasts such as myofibroblasts were only present in tumor tissues. Contrasting tumor vs normal analysis within the EC subsets we found that 48% of DE genes (e.g., *ANGPT2*) were common to analysis of both EC subsets. Other genes were potentially more EC subset specific (e.g., *ADAMTS18* is only expressed in tumor associated stalk-like ECs). (See **Supplementary Figure S8**).

A small number of DE genes may be attributed to ambient RNA from unrelated cell types including *CYP17A1*, which is highly expressed in adrenocortical cells (See **Supplementary Figure S8c**). As NAM tissues had abundant adrenocortical cells, we were careful to check the specificity of expression of specific DE genes in the manuscript when contrasting tumor and normal tissues. However, the number of DE genes that are clearly linked to ambient RNA appeared to be the minority and we do not think this can explain the distinct tumor and normal UMAP clustering patterns observed among EC and fibroblast cell types.

Third, still regarding the fibroblast and endothelial sub-clustering results, there is heterogeneity even within the same cell subtype of the same tissue compartment. Specifically, one can see that the "Stalk-like ECs" form two clusters in the tumor endothelial space. This type of unusual clustering pattern needs further investigation from the authors. If this variation turns out to be due to batch effect or other non-biological confounding factors, the authors need to revise the entire clustering strategy and take extra care when interpreting these clustering results.

Author response: We cannot easily explain the clustering of multiple stalk-like ECs. Firstly, it must be remembered that the cell classification was based on orthogonal cell classification using two other single cell RNA-seq cancer datasets and that cell types in PCPG tumors may not be entirely equivalent to other cancers.

We do not think the sub-clustering of stalk-like ECs is due to batch effect as we found that these cells arose from unrelated samples and different experimental batches (see **Supplementary Figure S8**). We attribute the clustering patterns to biological variability across cancers and individuals, although we cannot exclude minor technical artefact relating to sample quality. We have attempted batch correction for snRNA-seq data, but found the true known biological signal was removed as we described in response to reviewer 1. The importance of the subclusters of stalk-like cells is unclear. However, we do not think our interpretation of the data is voided by the presence of multiple sub-clusters of stalk-like cells in the tumors.

3. The analyses on the immune compartment of PCPG are lackluster and unreliable. Importantly, the major premises of the entire section that “Immune cell infiltrates in PCPG are predominantly macrophages” (line 267) and that “a low lymphocytic infiltrate is consistent with PCPG being immunologically cold” (line 299) are in direct conflict with a previous scRNA-seq study on PCs (PMID: 34905486) in which the most abundant cell types are in fact of the T cell lineage, followed by endothelial cells, myeloid cells, fibroblasts, and others. A widely appreciated drawback of snRNA-seq in comparison to scRNA-seq is that the former often subsamples T cells while retaining comparable myeloid abundance. Thus, without independent validation from an orthogonal data mode derived from the same set of samples (e.g. IHC, CyTOF, etc.) the authors simply cannot claim either that the PCPG microenvironment is dominated by macrophages or short of immune infiltration.

Author response: We have performed IHC analysis and validated that macrophages are more abundant than T cells in PCPG. CD3 IHC staining in 12 PCPG confirmed the relatively low number of T cells in tumors (mean = 5 cells/mm²) compared to CD163+ macrophages (mean = 105 cells/mm²) (**Figure 4f, Table S10**).

Additionally, the authors stayed on the very surface when analyzing the immune compartment of PCPG in terms of failing to reveal specific and often times well-known immune cell states and their interactions with PCPG subtypes or other disease features. The most conspicuous case supporting this argument is that the authors did not show any cell state diversity within the macrophage population and only mentioned a few M1 or M2 gene markers that seem to be differentially expressed between tumor and normal.

Author response: Additional macrophage M1/M2 genes have been added to the snRNA-seq analysis and this has been addressed in response to reviewer 3.

This problem also applies to the analysis of the T cell population where the authors only managed to annotate cytotoxic CD8+ T cells out of the entire CD8+ T cell population and Treg

out of the entire CD4+ T cell population, which is clearly not up to the current standard of single-cell immune analysis.

Author response: Given the paucity of T cells PCPG this precludes a finer analysis of T cell states.

Lastly, the authors failed to interpret appropriately what the differential expressions of certain immune cell markers actually mean in the context of bulk samples (similarly in figure 3g). For example, it seems that all four macrophage markers (CD163, MARCO, PLAU, and CXCL2) are more abundant in C1A2 compared to the other subtypes. But does it mean that C1A2 is of a more immuno-suppressive myeloid scenario or just that there are overall more macrophages in C1A2?

Author response: We presume that there are likely to be elevated number of macrophages in the C1A2 group. Although this conclusion was not supported by the snRNA-seq data, and we did not have sufficient archival material available to perform for IHC of C1A2 tumors there was higher expression of macrophage markers in the C1A2 tumors in bulk-tissue gene-expression data.

4. The issues described above, such as lack of formal statistical testing, arbitrary selection of gene markers as proxy of cell states/populations, failure in disentangling cell type abundance and cell type specific gene expression in analyzing bulk samples, all apply to the fifth section (“Schwann-cell-like cells (SCLCs) and putative paracrine signaling with NEO cells”) as well.

Author response: We have interpreted statistically significant DE genes based on association with cell type, biological process as well as potential relevance to PCPG pathology. We also provide extensive Supplementary data including full lists of genes in any analysis as described in the Supplementary Tables. We have also used gene set analysis incorporating published data where appropriate. Statistical tests and thresholds for P-values and log-fold change are now described in the manuscript where appropriate.

5. Although the authors were able to show that GPR139 expression is restricted to C1 PCPG, it may not bear a strong potential for PCPG therapeutics as anticipated by the authors. This is mainly due to the fact that GPR139 RNA expression is observable in fetal and adult neurons (see GTEX, Descartes, and Human Protein Atlas) and that GPR139 shows high antibody staining in normal adrenal gland granular cells, which makes it a non-exclusive marker of PCPG tumor cells.

Author response: As the reviewer noted, *GPR139* had a relatively restricted pattern of gene-expression in normal adult and fetal adrenal tissues with its expression largely restricted to normal and cancerous cells of the central nervous system. *GPR139* expression in normal cell types does not necessarily limit its application as a theranostic target. For example, *SSTR2* is a well-known theranostic target for neuroendocrine neoplasms. *SSTR2* is expressed by normal neuroendocrine cells but also lymphocytes (PMID: 11472428). Peptide receptor radionuclide therapy targeting *SSTR2* is a well-tolerated and effective radiopharmaceutical for treatment of PCPG (PMID: 28605448).

Minor comments:

1. *Eight PCPG gene-expression subtypes were determined in the analysis besides the normal subtype, but the authors claimed only seven in the abstract.*

Author response: As stated in the results (p7 line 184) two consensus clusters were combined as they coalesced, and the tumors had similar kinase signalling gene mutations consistent with the known C2A subtype. C2C was later excluded as it represented a false subtype driven by contaminating adrenocortical cell contamination leaving a total of seven “true” subtypes now described in the abstract.

2. *The term SCLC first appeared in figure 1d while its definition was given later in line 225.*

Author response: The call out and description of SCLCs has now been added to first section of the results (p5 line 142).

3. *No statistical tests were performed for analyses matching figures 5a-b, thus the claims regarding differential cell type abundance between subtypes are invalid.*

Author response: There was a mean difference in SCLCs between C1 and C2 subtypes, but a T test showed this was not statistically significant therefore we stated this in the manuscript (p11 line 335).

4. *There is a typo in figure 1h: the y axis should be “dim 2” instead of “dim 1”.*

Author response: The MDS analysis and figure has been removed from the manuscript

5. *There is a typo in Table S2: Fibroblast > Fibroblasts. Also, it seems what is named as “Smooth muscle cells” has taken the other name of “ACTA2pos fibroblasts” in this table. The authors need to adopt a unified naming scheme.*

Author response: Thank you for that observation. We have relabelled these fibroblasts as ACTA2+ fibroblasts in figures and tables.

6. *Why did the authors say “Two consensus clusters were consolidated...” (line 173) while there are nine in figure 2c?*

Author response: As noted above two of nine clusters from the consensus clustering were merged to make a single group C2A (Kinase). The C2C cluster was later removed as well as it was defined by adrenal cortex contamination.

7. *There is a typo in the title of the third section: remove the indefinite article “a”.*

Author response: the “a” has been removed from the third Results title.

8. Human reference genome hg19 is outdated. Mapping reads onto this genome is not the best practice although it is not expected to produce highly skewed results compared to mapping onto hg38.

Author response: Alignment to HG19 reflects the long duration of this project. We feel that a reanalysis using build HG38 would be an unreasonable expectation.

REVIEWERS' COMMENTS

Reviewer #1 (Remarks to the Author):

accepted

Reviewer #2 (Remarks to the Author):

The authors have elaborated on the points of concern to the point possible with the available data, and have provided additional clarity on other points of concern.

I have no additional remarks.

Reviewer #3 (Remarks to the Author):

The authors have satisfactorily addressed the points previously raised.

Reviewer #4 (Remarks to the Author):

The authors have offered a comprehensive response to the reviewer's feedback, addressing the majority of concerns and offering new experimental data to support the findings. I would like to thank the authors for addressing my comments thoroughly and I think the paper is appropriate for publication in its current form.